# A framework for visualizing and describing city image promotion short video data based on microcube model

**Jing He** [1,2]*, **Zhuoluo Yang**[3], **Jiayi Zhu**[1]

**1** Institute for Advanced Studies in Humanities and Social Science, Beihang University, Beijing, China, **2** Guangxi Key Lab of Multi-source Information Mining & Security, Guangxi Normal University, Guilin, China, **3** State Key Laboratory of Media Convergence and Communication, Communication University of China, Beijing, China

\* 18610560816@163.com

## Abstract

### Background

The rapid development of media technology and media environment provides rich resources and convenient ways to shape the image of the city. Short video has become an important help to shape the image of the city and build the city brand. How to use short video to shape the image of the city is the key link of urban construction. This study focuses on six primary variables for analyzing city image short videos: unexpected events, emotional resonance, scene transition, elemental amplification, element interaction, and screen style. These variables were selected based on their demonstrated impact on short video engagement and dissemination efficiency.

### How

In order to realize comprehensive analysis of short video content, This study collected 20,668 video screenshots from the Douyin platform as data samples. Data collection spanned July 2019 to December 2019, and analysis was conducted using the Python programming language with the Pandas and Matplotlib libraries for data processing and visualization.

### Objective

To reveal the relationship between video content features and popularity by quantitative and visual methods, and to provide reference for optimizing urban brand promotion strategies.

### Conclusion

(1) The microcube model shows strong flexibility and applicability in short video content analysis, and can help reveal the complex relationship between short video content characteristics and communication effect. (2) Unexpected events and emotional resonance are key factors in the attractiveness and communication effect of short videos. Reasonable design of scene switching and element interaction significantly enhances the visual impact of short videos.

**Data availability statement:** All relevant data is available on: https://zenodo.org/records/14633615.

**Funding:** Supported by Key Laboratory of Spatial Data Mining & Information Sharing of Ministry of Education, Fuzhou University (No.2023LSDMIS02); Research Fund of Guangxi Key Lab of Multi-source Information Mining & Security (No.MIMS22-11); Anhui Cultural Tourism Innovative Development Research Institute (No.ACTK2022YB01). The funders conducted a compliance review of the manuscript.

**Competing interests:** The authors have declared that no competing interests exist.

## 1 Introduction

City image promotion has become an important issue in city management and economic development. Especially in today's rapid development of digital media, short video stands out among many communication methods in the field of new media, combining the "mirror" of the city in the form of fragmentation, uniqueness and fluidity, and rapidly spawning a variety of new Internet celebrity cities, which has great influence in attracting potential tourists, and also provides an important platform for the export of urban culture and international brand image. However, the performance of short video propaganda is obviously different in different cities, and there is a lack of systematic methods to evaluate the specific impact of video content on the visibility and communication effect of cities. Therefore, how to analyze short video content from the perspective of data and visualization and provide optimization suggestions for urban publicity strategies has become an important issue that needs to be explored. Most of the existing researches focus on the communication characteristics or creative skills of short video, and lack of quantitative content analysis to analyze the specific role of short video in shaping city image. For example, how to analyze short video content more scientifically and extract the key factors affecting the dissemination effect? Starting from micro perspective, this study proposes a short video analysis framework to explore the correlation between short video content features and communication effects. This analysis framework can not only provide ideas for the brand communication of Chinese cities, but also provide reference for the optimization practice of urban image publicity under the background of globalization.

## 2 Related work

At present, domestic and foreign scholars mainly focus on the content production, communication characteristics, development status and ethical dilemmas of short videos, and relatively little research is conducted based on the screen perspective of short videos. After reviewing and summarizing the existing related literature, it is found that short video screen research mainly focuses on six types.

(1) Unexpected events: Defined as unpredictable, attention-grabbing moments in video content that enhance viewer engagement. Huang Feng [1] suggests that the unexpected events in life are irreproducible and will be presented in the short video screen to become a great highlight and gain more attention from the audience. Qin Feng et al. [2] found that differentiated video material, ups and downs, and innovative elements would reduce viewers' boredom by analyzing short video user comments. M Li et al. [3] pointed out that content involving unexpected scenes (such as emergencies or unexpected turns) in videos has a much higher forwarding volume than that of ordinary videos. This "surprise effect" suggests that unexpected events play a significant role in attracting attention. Wang et al. pointed out that the introduction of unexpected events in short videos not only increases the audience's attention, but also improves the breadth and depth of information dissemination. The empirical study of Chen et al. shows that the rational use of emergencies in short videos can effectively attract the attention of young users and improve their willingness to interact. The empirical study of Ren J et al. [4] shows that the rational use of emergencies in short videos can effectively attract the attention of young users and improve their willingness to interact.

(2) Emotional resonance: Described as the ability of video content to evoke emotions, creating a deeper connection with the audience. Cheng Qianqian [5] proposes that adding regional cultural elements of revolutionary base areas such as dialects in short videos of red documents can create a familiar cultural atmosphere for viewers, reinforce the sense of regional belonging while restoring history, and enhance viewers' immersion and sense

of immersion. Huang Xiaoyin [6] argues that emotion is currently recognized as the basis for sustaining users and is the strongest link between users and the platform, whether it can inspire users' emotional resonance has also become an important indicator to judge whether media products have influence. As the head IP of food short videos, "Eclipse", uses the cooking process of food to convey the attitude of "living well" and inspires users to identify with and "be in" the emotional level by constructing a specific daily life space [7]. This will lead to the emotional resonance of users. B Wang et al. [8] found that short videos can significantly increase users' willingness to share by enhancing positive emotional expression, especially in content involving cultural identity and community belonging. Pang N et al. [9] found in the analysis of emotional elements of short video communication that emotional resonance plays a key role in improving user engagement and secondary communication effect.

(3) Scene transition: Refers to seamless or dynamic shifts between video scenes that maintain visual continuity and narrative flow [10]. According to Sun Zhenhu [11], the reasonable use of transitions can make the content presentation rational and the story elaboration clear, so that the audience can easily understand and accept it. Yan Hanmei [12] introduces three types of techniques to achieve transitions from the perspective of screen editing: fluent editing, matching editing and jump cutting, and selects appropriate editing techniques to meet the best viewing experience of viewers according to different short video production needs. The reasonable use of transitions can also make the news report free from traditional forms. For example, the short video of "One Day Patrol of Civil River Chief" presents the whole "river chief system" activities in a minute with multiple scenes, transitions and angles [13]. Y Zhou [14] proposed that rapid scene switching can enhance the rhythm of the video, but if the switching is too frequent, it may reduce the viewing comfort of the audience. Chen S et al. [15] proposed that scene transformation is closely related to narrative rhythm, and reasonable scene cohesion can avoid information loss to the greatest extent and ensure content integrity.

(4) Elemental amplification: Highlights specific details or elements in the video to enhance content memory [16]. After studying Breakfast China, Fan Yuanyuan [17] found that the short documentary left behind large scenes, used more close ups and close-ups to show people and actions, paid attention to detail control, and presented breakfast such as steaming Fujian Saiki fried buns and fragrant Guangdong Chaoshan pig's blood soup on the small screen to truly show people's living conditions. The short video "Tribute to the People's Police" produced by Quanzhou Radio and Television Station in conjunction with Quanzhou Public Security Bureau contains close-up images of public security police using DNA technology for data comparison and analysis and research, highlighting the professionalism of the people's police and how technology can help criminal investigation work [18]. Y Lin [19] analyzed close-ups in short videos and found that enlarging detailed elements (such as food texture and architectural decoration) can effectively attract viewers' visual attention and enhance content memory. Yang Y et al. [20] emphasized that the amplification of detail elements plays a key role in enhancing the audience's content memory and brand recognition.

(5) Element interaction: Refers to the dynamic relationships and movements between visual elements in a video [21]. For example, the characters in "The Day of Food" include a Persian cat endowed with a distinct personality in addition to the chef Jiang Laodao, who is the introducer at the beginning of the video, the spectator of the cooking process, and sometimes the taster of the final result. The short video does not set up human emotions, but expresses emotions from the cat's point of view, making the video easy and interesting, but real and natural [22]. Shen Yang [23] argues that in short video refinement

production, the small depth-of-field images captured by the combination of large aperture and telephoto can obtain a sharper perspective and reflect the image texture and ingenuity as it is. MY Shedid [24] found that in short videos, dynamic interactive scenes (such as the dialogue between natural elements and modern buildings) can enhance the audience's immersion through the integration of multi-modal information. Sun G M et al. [25] found that the interactive relationship of visual elements plays a bridging role in information-transmission and audience emotional connection.

(6) Screen style: Describes the overall visual tone, atmosphere, and aesthetic choices in short videos. Hao Xiaoyuan [26] took One, Ershang and Pear Video as the research objects and found that the styles of One and Ershang are both inclined to fresh and literary style, and Pear Video presents a variety of styles due to its worldwide patron system. Shen Qingbin [27] found that most of NHK's documentaries use plain means to pick up images and focus on the communication of live atmosphere. Compared with the BBC, where every shot is perfect, NHK's documentary images are more plain and simple, and low-key and subtle, so he believes that NHK's documentary images as a whole present a rustic style. In the urban documentary Jin Zhi Yang [28], subdued tones are used to adjust the relation-ship between light and shadow to outline the scene of the dignified and mysterious urban architectural complex of Taiyuan, Shanxi Province, highlighting the unique historical and humanistic atmosphere of Taiyuan. L Ma [29] analyzed the picture styles in urban short videos and found that highly consistent and aesthetically rich visual styles are more likely to establish audience's emotional attachment to video content. For example, fresh-styled videos are significantly more appealing to younger users than other genres. Peng L [30] found that tone and visual style play a significant role in short video narration and can significantly affect the viewing experience of the audience.

In addition to the above six parts, scholars at home and abroad have also studied other aspects of picture content, such as multimedia technology, legal copyright, post-production, shooting composition, etc., but literature data are even scarcer. For example, Cui Guobin [31] explored whether live sports event screens have originality in the sense of film works. Peng [32] and others introduced the latest progress in video coding of screen content. Wang Zhen [33], on the other hand, analyzed the principles that should be followed in filming current affairs news TV screens and discussed the filming techniques of different types of TV current affairs news. Zhang Jieliang et al. [34] explored the method of video conference screen quality evaluation from both subjective and objective aspects.

In summary, this paper obtains city image short video data with geographical location mark-ers, takes the research literature of domestic and foreign research scholars on short video images as a reference basis, combines the presentation patterns of city image propaganda on short video, including content themes, emotional bias, media sources, communication characteristics and dynamic assessment of overseas media, traditional mainstream media communication dis-course insight, official media and local self-media synergistic development The content produc-tion characteristics of short video images are extracted and initially set as six imagery elements: unexpected events, emotional resonance, scene transformation, element amplification, element interaction, and picture style, as the research dimensions of the data visualization description framework of city image propaganda short video based on the microcube model.

## 3  Microcube model framework construction

The Micro-Cube Model proposed in this study aims to provide a quantitative and visual analytical framework for understanding the relationship between short video content char-acteristics and their effectiveness in urban image promotion. Below, we present a detailed

explanation of the model's construction, covering data sources, data processing, model building, visualization, reproducibility, and applicability, ensuring a clear, transparent, and replicable methodology.

(1) Data Sources: The dataset for this study was sourced from the Douyin platform, focusing on short videos explicitly aimed at urban image promotion, ensuring a balanced representation of both popular and less popular cities. Selection criteria included: videos must have clear geographic tags and center on urban branding themes; invalid or noisy data, including privacy violations, were excluded; all data collection complied with platform policies and ethical guidelines.

(2) Data Processing: The collected data were processed using Python programming language, with Pandas and Matplotlib libraries applied for data cleaning, organization, and visualization. The video timeline was segmented into micro time units, where each frame was treated as an independent data unit. These frames were then mapped into Micro-Cubes, representing granular units for time-based and spatial content analysis. Each Micro-Cube was further deconstructed into visual elements (e.g., technical elements, natural landscapes, urban imagery). A binary encoding system (1 for presence, 0 for absence) was used to categorize and quantify these elements. Visual elements were then color-coded and displayed along a time-series axis, forming a continuous visual representation of content distribution and variation over time.

(3) Model Building: The Micro-Cube Model integrates six core visual elements—Unexpected Events, Emotional Resonance, Scene Transition, Elemental Amplification, Element Interaction, and Screen Style—each of which was embedded into the model for systematic analysis.

- Unexpected Events: Frames with significant anomalies or content jumps were marked as "unexpected events" using threshold-based anomaly detection. Heatmaps and time-series graphs visualized their distribution.

- Emotional Resonance: Emotions were classified into nine categories (e.g., joy, sadness, surprise) using RGB color encoding grids. Emotional patterns were matched to each time unit and visualized across time grids.

- Scene Transition: The video timeline was segmented into narrative time and physical space, with transitions identified based on significant shifts between frames. Color-block transition maps displayed frequency and spatial movement.

- Elemental Amplification: Prominent visual elements occupying over 30% of the screen for more than 3 seconds were tagged and color-coded for visual emphasis.

- Element Interaction: Relationships between visual elements (e.g., overlaps, narrative associations) were analyzed, decomposed into sub-elements, and dynamically visualized using interaction heatmaps.

- Screen Style: The primary visual tone (e.g., bright, dark, neutral) was extracted from each frame and displayed using tonal distribution maps.

(4) Implementation Process: The construction and implementation of the Micro-Cube Model followed a sequential workflow:

- Time Segmentation: The video timeline was segmented into micro time units.

- Micro-Cube Mapping: Each time unit was mapped into a Micro-Cube, with visual elements encoded accordingly.

- Element Encoding: Visual elements were categorized and encoded using binary values.

- Data Visualization: The encoded data were visualized using time-series graphs, heatmaps, and interaction maps.

- Statistical Analysis: The distribution and frequency of visual elements were statistically analyzed, and their correlation with audience engagement metrics (e.g., likes, shares, comments) was assessed.

- Reproducibility and Applicability: The Micro-Cube Model was designed with transparency and reproducibility as key priorities. Detailed segmentation rules, encoding standards, and visualization methods ensure that other researchers can replicate the model using similar datasets and tools. Beyond urban image promotion, the model is adaptable to other fields, including brand communication, cultural content dissemination, and user behavior analysis, showcasing its scalability and versatility.

## 3.1 Unexpected events - sense of novelty

Taking the micro perspective as the analysis perspective, the video content is refined and a content microcube model is established to encapsulate the complete short video into multiple fragmented short videos according to the time interval, i.e., each small cube encapsulates the video content per unit of time, as shown in Fig 1(a).

Then an assignment operation is performed on each small cube. This operation contains multiple inputs a1, a2......an (technology elements, food elements, natural landscape elements, cityscape elements, people elements, animal elements, and other imagery elements), an output z, and multiple intermediate parameters (the number of parameters is comparable to the number of time intervals). Among them, a1, a2......an are set to exist equal to 1 and not exist equal to 0. The intermediate parameter values take values that vary in multiples. A directional arrow indicating the connection can be understood as follows: at the input side, the size of the transmitted signal is a, and there is a change parameter w in the middle of the end, and the signal becomes a＊w after several changes.

Let the total duration of the short video be t. After n times of segmentation, the signal size at the output end becomes $a*\left(\dfrac{w_1}{n} + \dfrac{w_2}{n} + \cdots + \dfrac{w_n}{n}\right)$, i.e., $a*\dfrac{w}{n}$. If the range of $\left|\dfrac{a*w}{n} - a\right|$

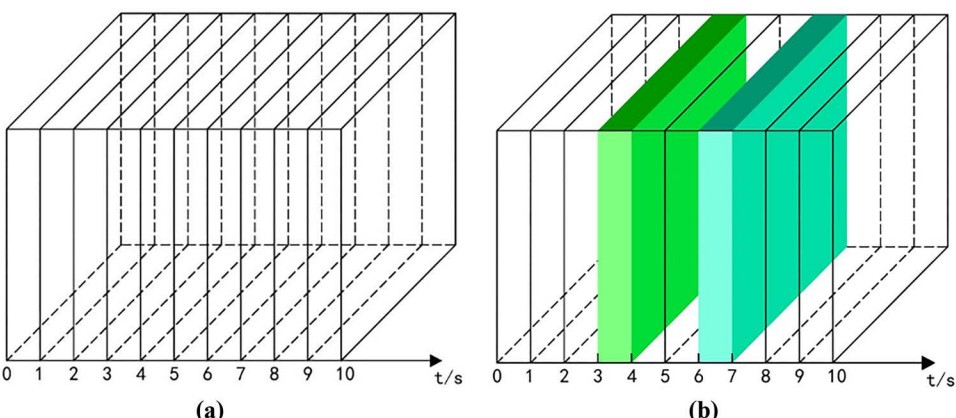

**Fig 1. Refinement analysis of the content presentation module.** (a) content cube cut. (b) accidental event microcube visualization model.

satisfies the threshold, the video content display is considered smooth. if the range exceeds the threshold, the video content display is considered transitive and contains the transitive element T. The event that triggers this transitive is called an unexpected event. Fig 1(b) shows the schematic diagram of the occurrence of unexpected events in short videos. 3-4s and 6-7s are both transitions time domain, and the transitions region is visualized by color, and its shades represent the degree of transitions.

## 3.2 Emotional resonance - sense of immersion

Short videos are in a sense emotional product in the process of communication. Experimentally, we designed an emotion grid chart with nine emotions including moved, praised, funny, happy, neutral, angry, sad, shocked and helpless, and used #EE6363, #EEDFCC, #FFA54F, #FFEC8B, #FFFFF0, #00F5FF, #63B8FF, #00FF7F and #CD69C9 as RGB color controls, respectively. as shown in Fig 2(a).

Then, the emotion grid map is color-matched with the content cube model (neutral to white, not identified as color system, and its fractional video number is set to N), and the result of matching is that the emotion marker is implemented to the content cube model, and the marker color is consistent with the RGB color type in the emotion grid map. Then the complete short video will be divided into multiple color-based fractional short videos A according to the time interval, i.e., each small cube encapsulates the emotional content per unit time. Among them, the color matching degree $M = \dfrac{A-N}{A}$, when $M > 30\%$, is strong in emotional substitution. Fig 2(b) is a schematic diagram of color matching degree in the city image video with $M = 50\%$, 2-4s, 7-8s, 8-9s, and three emotions of happy, shocked, and helpless, respectively.

## 3.3 Scene transition - sense of following

Scene transitions imply the existence of a relatively large change in temporal or spatial relationships. Time is divided into broadcast time, event time, and narrative time. space is divided into physical space and psychological space. physical space is the display space of actual transitions, and psychological space is the space adjusted with the ups and downs of the storyline. The TS level transition scenes include T1, T2, T3, S1, S2 and Tn/Sn. Among them, T1, T2, and T3

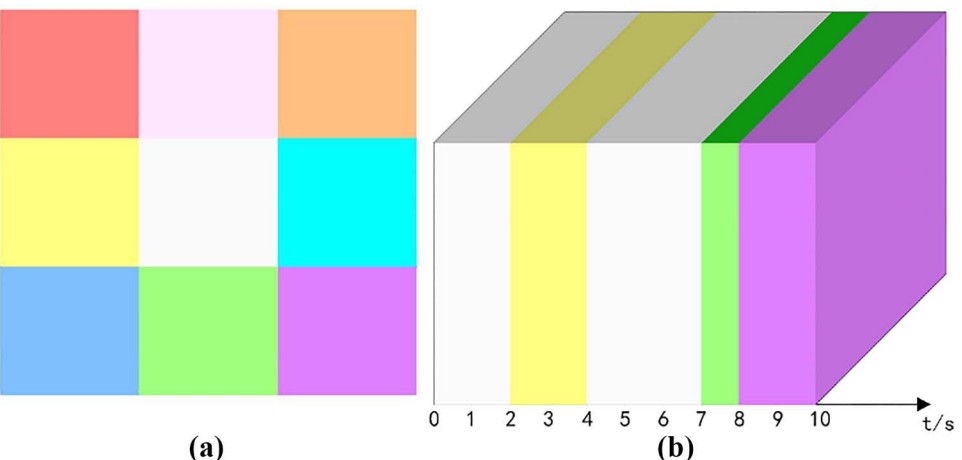

**(a)** **(b)**

**Fig 2. Refinement analysis of emotional carryover.** (a) emotional grid diagram. (b) color matching results.

represent the corresponding broadcast time conversion, event time conversion, and narrative time conversion. S1 and S2 represent the corresponding physical space conversion and mental space conversion. and Tn/Sn represents the hybrid conversion of time and space connection. The TS level conversion domain model describes the abstract static structure of the system, and only after matching the dynamic behavior can the complete representation of the system be completed.

In the TS level conversion scenario, two typical metrics, switching correctness and real-time, are selected. Switching correctness requires the following 2 conditions to be discerned: first, based on the completion of the short video to be able to enter successfully different scene states. second, traversing the frame state paths of all standard samples in the short video. Real-time metrics require not only the complete short video playback to respond within a determined time, but also to ensure the correctness of the response. In this scenario, once the relevant time constraint is violated, it will not only affect the efficiency of scene transition, but may also cause the coherence degree of video playback.

Analysis of scene conversion for the video after content refinement: let the total propagation time of the short video be t. After n times of segmentation, visualize each scene as a different color cube within the unit time, and the color switching represents the conversion of the scene, and let the number of scene switching be q. In Fig 3(a) ~ 3(d), q is equal to 3, 2, 0, and 6, respectively.

## 3.4 Elemental amplification - sense of in-depth

In fact, for the short video platform, even the city image promotion is shown as "content production", which usually contains one or more imagery elements such as technology elements, food elements, natural landscape elements, urban landscape elements, people elements, animal elements, etc. We designed the imagery elements grid and used #00FF7F, #FFEC8B, #EE6363, #00F5FF, #EE82EE, #FFA54F as RGB color controls, as shown in Fig 4(a).

The video content is further refined to establish the tiny content cube model, then each fragmented short video is further divided into 100 sub-cubes equally, i.e., each sub-cube encapsulates the video content per unit time and per unit screen, as shown in Fig 4(b). Let each sub-cube be 1 unit, when cube cutting is performed on the short video sample data, one or more of the imagery elements may appear on the video screen per unit of time. The percentage of occupied screens for each imagery element is quantitatively analyzed and visualized with different colors. Elements with screen occupancy greater than 30% and scene

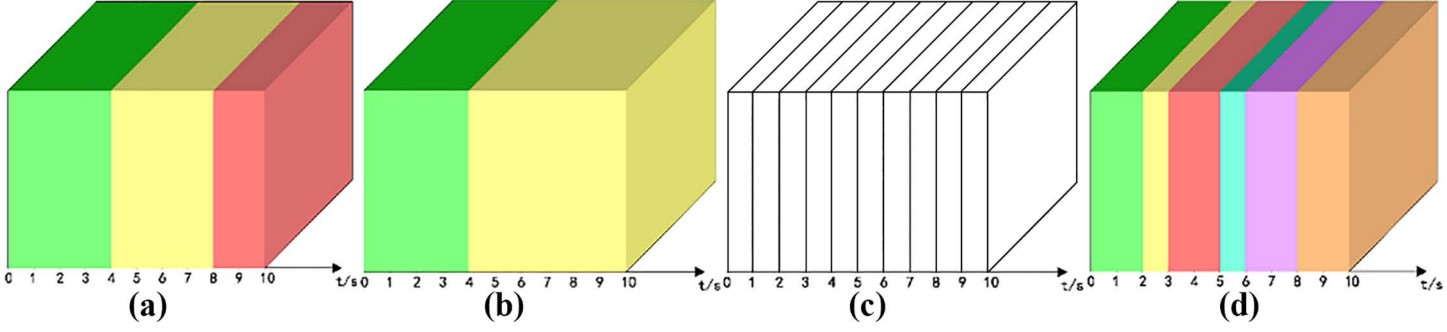

**Fig 3. Scene transition microcube visualization model** (a)scene transitioned three times. (b) scene transitioned twice. (c) no scene transition. (d) scene transitioned six times.

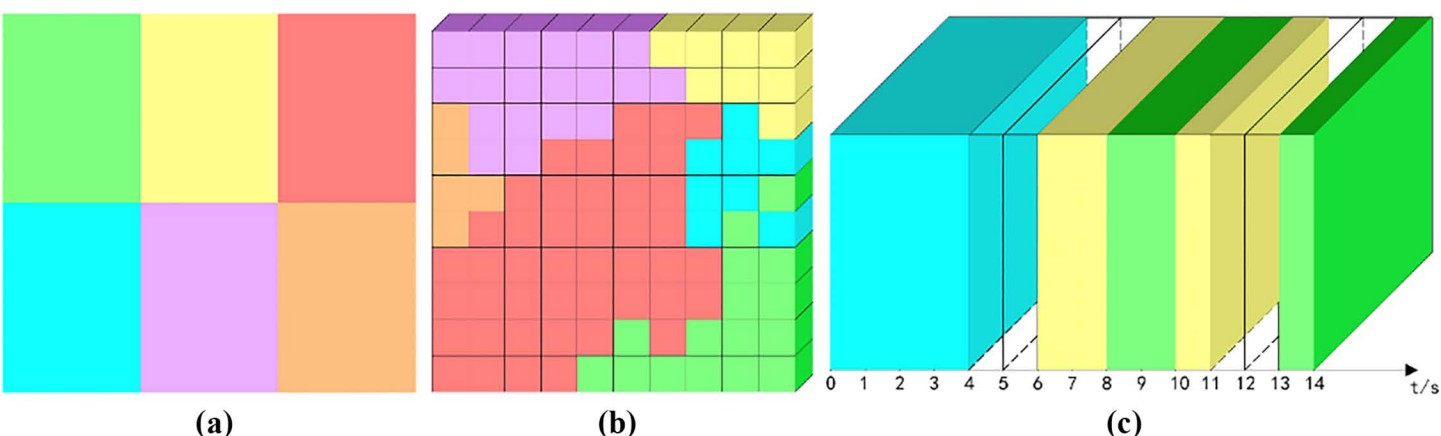

**Fig 4. Refinement analysis of element amplification** (a) color grid diagram of imagery elements. (b) sub-cube model. (c) this sample data contains 4 amplified elements.

time greater than 3s are extracted and identified as amplified elements, set as E. Elemental amplification is relative to each other and there is no gradual change over time or other measure. Representation = f(e1, e2,., e3), where e is the elements that characterize the short video content. Fig 4(c) shows four amplification elements, where 0-4s are cityscape elements, 6-8s and 10-11s are food elements, and 8-9s and 13-14s are technology elements.

## 3.5 Element interaction - sense of dynamism

The analysis of the amplified elements continues with further decomposition of elements of the same category. The decomposition is based on the principle of interaction, i.e., when there is interaction between elements of the same category, only then the element continues to be decomposed into multi-category interactive elements, and the original element is represented by OE and the decomposed element is represented by DE. The visualization color of DE is determined by the visualization color of OE as a sub-color family of OE. Let the total propagation time of the short video be t. After n times of segmentation, the microcubes with different color particles are visualized according to the element categories in each subcube per unit time, among which, the original element of technology OE technology is original green, the original element of food OE food is original yellow, the original element of natural landscape OE natural landscape is original red, the original element of urban landscape OE city is original cyan, and the character The original element OE characters is the original purple, and the original element OE animals is the original orange. When the imagery element is judged to be an interactive element, OE technology is decomposed into OE technology 1 + OE technology 2, OE cuisine is decomposed into OE cuisine 1 + OE cuisine 2, OE natural landscape is decomposed into OE natural landscape 1 + OE natural landscape 2, OE cityscape is decomposed into OE cityscape 1 + OE cityscape 2, OE character is decomposed into OE character 1 + OE character 2, and OE animal is decomposed into OE animal1 + OE animal2, as shown in Fig 5(a).

Let the total propagation time of the short video be t. After n times of segmentation, the different sub-cubes per unit time are decomposed into multiple micro-cubes with different colors, as shown in Fig 5(b). Firstly, the amplified elements are judged according to the method described in 4.4, and then the decomposition analysis of the elements is performed on the amplified elements, and the color decomposition is recorded, and Fig 5(c) demonstrates the decomposition rate d = 70%.

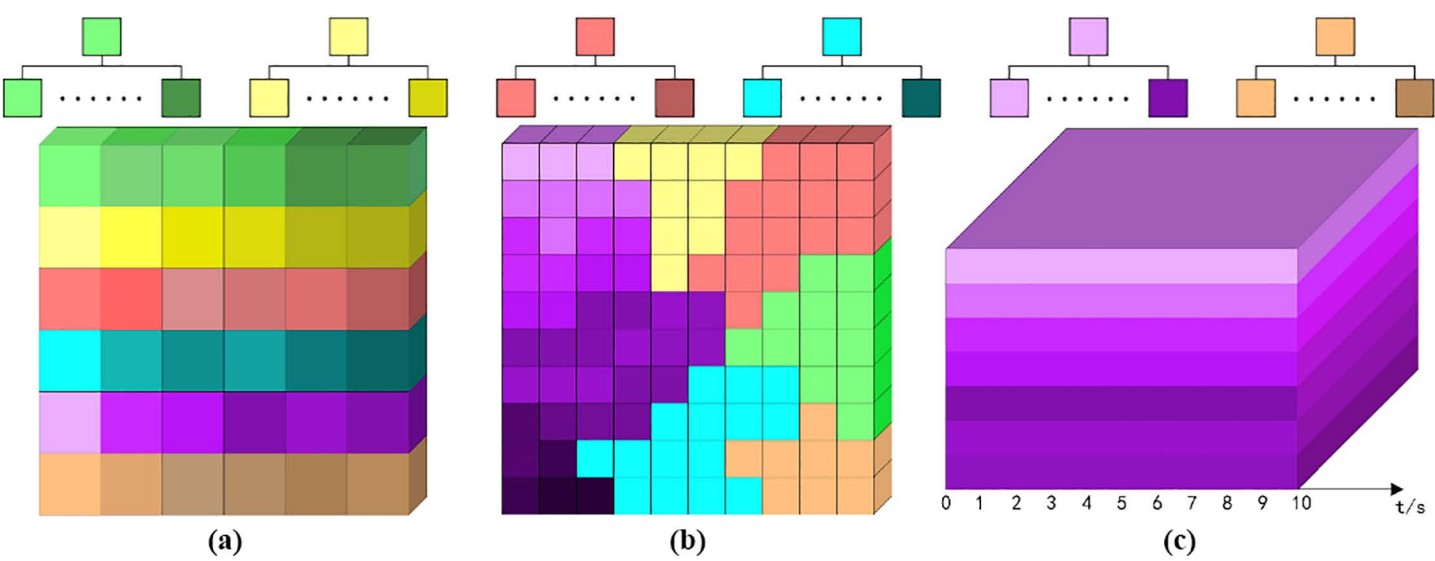

**Fig 5. Refinement analysis of element interactions** (a) color grid diagram of the decomposition of imagery elements. (b) microcube model of character element interactions. (c) decomposition of 7 interactions of character elements.

### 3.6 Screen style - main color

Each image in the video can be abstracted into three forms: point, line and surface, and the effective presentation of these forms are the elements. The above decomposition of different elements has been carried out, and the decomposed elements are the basic framework to form the microcube model. However, the video screen is often more than a simple combination of individual elements, and sometimes involves an effective merging of elements to form the style of the overall video screen. A grid diagram of the main colors of the screen style was designed, and #0000FF, #BEBEBE, #FFFACD, #6495ED, and #FFC0CB were used as RGB color controls for dark, gray, light, medium, and bright tones, respectively, as shown in Fig 6(a). Fig 6(b) shows five main colors of the screen, among which, 0-2s and 4-5s are bright tones, 2-4s are gray tones, 5-7s are dark tones, 7-8s are light tones, and 8-10s are medium tones.

## 4 Analysis of cases

There are three levels in the short video communication effect rating index model: the criterion level, the sub-criterion level and the solution level. The criterion layer is based on the hotness, breadth and depth of communication. the sub-criteria layer sets the second-level indicators based on the media hotness, fan hotness, search hotness, reputation of each network platform, content acceptance, cross-domain linkage, crowd reach, thematic distinctness and thematic diversity. the solution layer corresponds to the sub-criteria layer and refines the relevant indicators. After applying the index model to evaluate the communication effect of the short videos of Beijing city image propaganda, the absolute hotness (H) concept is used to calculate the hotness value of the short videos of online celebrities using a one-year time period as the time period of attention: $H = (xC_a + yC_b + zC_c)(1 + C_d)$. Where H denotes the absolute hotness value. $C_a$ denotes the value calculated by the propagation hotness criterion layer. $C_b$ denotes the value calculated by the propagation breadth criterion layer. $C_c$ denotes the value calculated by the propagation breadth criterion layer. $C_d$ denotes the account quality of the video publisher (depending on the account quality of the video publisher, the value is between 0 and 1). x and y denote the factor weights (depending on the importance of the

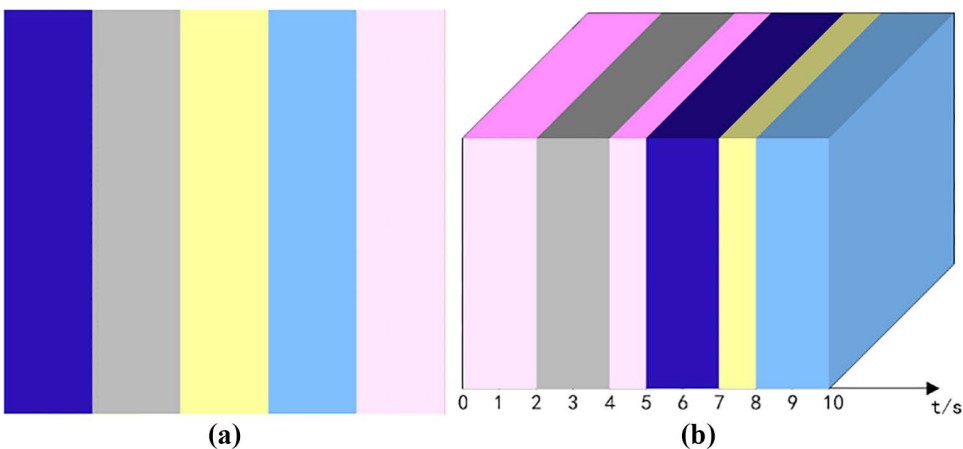

**Fig 6. Refinement analysis of picture styles** (a) color grid diagram of primary colors. (b) visual framework of 5 primary colors.

factor, the value is between 0 and 1).. Accordingly, the city image promotion short videos in different heat ranges are categorized. Appendix I shows the design model of short video hotness index system.

Based on the hotness value of the short videos of Netflix, this experiment selected a total of 1200 short videos of popular city image propaganda short videos from the TikTok platform with likes> 1w and retweets> 0.8w and non-popular city image propaganda short videos with likes < 1w and retweets < 0.8w within the second half of 2019, and a total of 20,668 video screenshots were obtained after the screen draw seconds per second, and ensure that the video content includes geolocation tags and a clear city promotion theme. To ensure the authenticity and representativeness of the video content, the research team manually reviewed all selected videos and cross-verified them with the metadata provided by the platform, such as upload date, tag and post account number, to eliminate possible noise data (In the obtained videos, all content is strictly in accordance with the data use policy, only for academic research, and does not involve personal privacy information). Table 1 shows the number of microcubes and the number of video screenshots per second in the framework of visualization description of city image promotion short video data based on microcubes model for popular cities and non-popular cities. Further, plot the relationship heat matrix for popular and non-popular cities in the following dimensions: the number of unexpected microcubes, number of unexpected frames, number of

**Table 1. Overview of the microcube model constructed for popular/non-popular cities.**

| popular cities | Beijing | Chengdu | Guang-zhou | Nan jing | Nan ning | Shanghai | Shen zhen | Wu han | Xian | Zhang jiajie | Chang sha | Chong qing | Total |
|---|---|---|---|---|---|---|---|---|---|---|---|---|---|
| Number of micro-cube models | 12 | 12 | 11 | 13 | 13 | 10 | 13 | 12 | 13 | 11 | 11 | 13 | 144 |
| Number of video frames in seconds | 216 | 205 | 205 | 232 | 216 | 149 | 220 | 186 | 210 | 154 | 182 | 218 | 2393 |
| non-popular cities | Alashan-meng | Bao ding | Bei jing | Chengdu | Fo shan | Guang-zhou | Hang zhou | Shanghai | Shen zhen | Xian | Zheng-zhou | Chongqing | Total |
| Number of micro-cube models | 13 | 13 | 9 | 7 | 11 | 10 | 11 | 10 | 10 | 9 | 10 | 10 | 123 |
| Number of video frames in seconds | 211 | 182 | 136 | 107 | 157 | 140 | 164 | 154 | 170 | 135 | 131 | 147 | 1834 |

emotion resonance microcubes, number of positive emotion microcubes, number of negative emotion microcubes, number of emotion resonance frames, number of positive emotion frames, number of negative emotion frames, number of scene transition microcubes, number of scene transition frames, number of elemental amplification microcubes, number of elemental amplification frames, number of microcubes with at least two amplification elements, number of frames with at least two amplification elements, number of elemental interaction microcubes, number of elemental interaction frames, number of gray tone microcubes, number of gray tone frames, number of dark tone microcubes, number of dark tone frames, number of bright-toned microcubes, number of bright-toned frames, number of mid-toned microcubes, number of mid-toned frames, number of light-toned microcubes, and number of light-toned frames. The relationship heat matrices are plotted in terms of the number of frames, the number of bright-tone microcube models, the number of bright-tone pumping frames, the number of mid-tone microcube models, the number of mid-tone pumping frames, the number of light-tone microcube models, and the number of light-tone pumping frames, as shown in Fig 7, each using the same emerald green color space, but using a different color mapping. The greener the color represents the stronger the correlation between the two factors, and is dark green when both are the same element (as shown in the diagonal of the matrix), indicating a 100% correlation. The redder the color means the less correlation between the two factors. Comparing the two heat matrix plots, it is found that there are significantly more green color blocks in popular cities (a) than in non-popular cities (b), and the green color concentration is greater. while there are significantly more red color blocks in non-popular cities (b) than in popular cities (a), "more than the number of microcube models of both elements" The number of "microcubes with more than two elements" is negatively correlated with nearly half of the other elements, and the number of brightly colored second frames is significantly negatively correlated with the

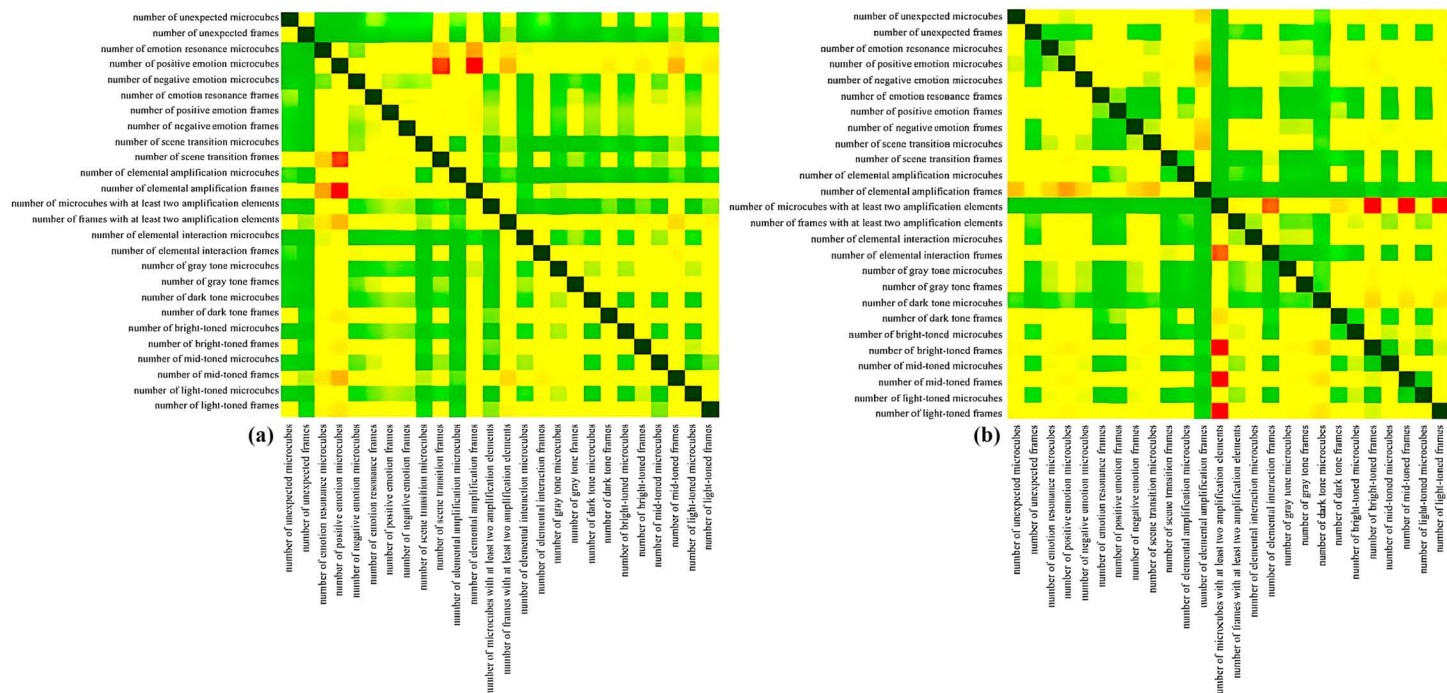

**Fig 7.** (a) Relationship heat matrix diagram of short video screen visualization relationship heat matrix display of sample data for popular cities. (b) relationship heat matrix display of sample data for non-popular cities.

number of brightly colored second frames. The above analysis shows that there is a correlation between the intention elements of short videos from popular cities, i.e., there may be multiple intention elements in the same picture, while the intention elements of short videos from non-popular cities have a weaker relationship with each other, and the intention elements in the video picture are relatively single.

Due to space limitation, only 144 city popular short video microcubes models with 2393 city popular short video screenshots and 123 city non-popular short video microcubes models with 1834 city non-popular short video screenshots are shown here. To clearly demonstrate that the microcube model-based descriptive framework for visualizing city image promotion short video data can be used to perform descriptive analysis of content abstraction of popular and non-popular city short videos occurring in a time series, a stack of microcube models is used for presentation.

## 4.1 Visualization result of unexpected events of short videos based on microcube model

Based on the display data, the number of microcube models containing unexpected events and the number of video pumping second frames were recorded as shown in the following Table 2. The number of microcube models containing element T in the popular city image videos is 106, and the number of video second draws is 408, while the number of microcube models containing element T in the non-popular city image videos is 39, and the number of video second draws is 123.

The visualization results of unexpected events of short videos based on microcube model are shown in Fig 8.

## 4.2 Visualization results of emotional resonance of short videos based on microcube model

Based on the display data, the number of microcube models containing the nine emotions and the number of video pumping second frames were recorded as shown in the Table 3.

The number of microcubes containing emotional elements in the urban popular short videos was 69, and the color matching degree of the total model $M1 = 47.92\%$. Among them, the number of microcubes involving positive emotions is 47, the color matching degree of the total model $M1 = 32.64\%$, the color matching degree of the total emotion $M2 = 61.84\%$. the number of microcubes involving negative emotions is 29, the color matching degree of the total model $M1 = 20.14\%$, the color matching degree of the total emotion $M2 = 38.16\%$. the number of microcubes involving both positive and negative emotions is 7. The number of microcubes involving both positive and negative emotions is 7, the color matching degree of total models $M1 = 4.86\%$, and the color matching degree of total emotions $M2 = 9.21\%$.

The number of microcubes containing emotional elements in the non-popular city image videos is 35, and the color matching degree of the total model is $M = 28.46\%$. Among them, the number of microcubes involving positive emotions is 32, the color matching degree of total models $M1 = 26.02\%$, the color matching degree of total emotions $M2 = 91.43\%$. the number of microcubes involving negative emotions is 5, the color matching degree of total models $M1 = 4.07\%$, the color matching degree of total emotions $M2 = 13.51\%$. the number of

**Table 2. Number of microcube models containing unexpected events and number of video pumping seconds frames.**

|  | Popular city image short video | Non-popular city image short video |
|---|---|---|
| Number of microcube models containing element T | 106 | 39 |
| Number of video frames in seconds | 408 | 123 |

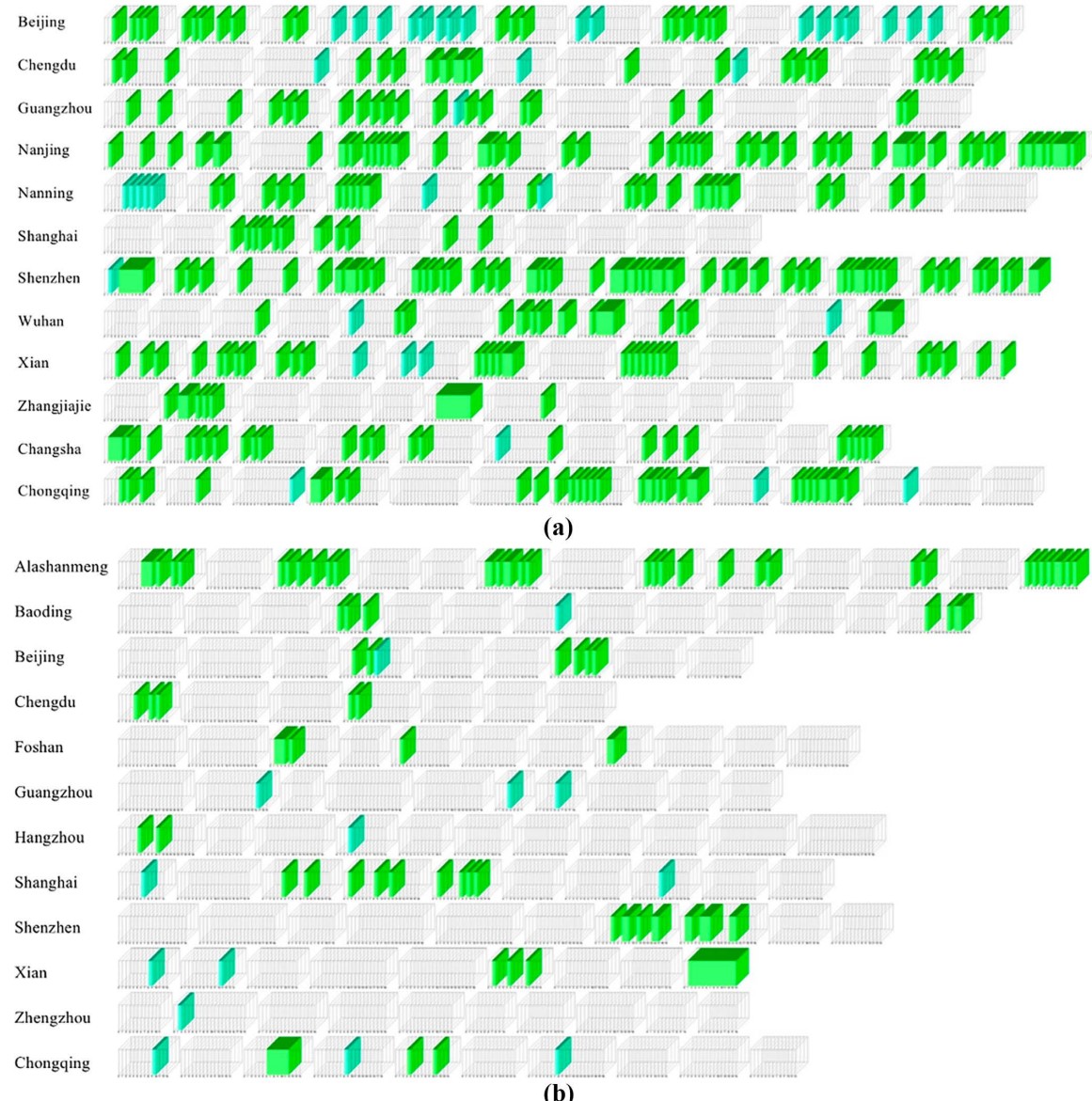

**Fig 8. Accidental event stacking graph based on microcube model** (a) sample case of popular city image short video. (b) sample case of non-popular city image short video.

microcubes involving both positive and negative emotions The number of microcubes involving both positive and negative emotions was 2, the color matching degree of the total model was M1 = 1.63%, and the color matching degree of the total emotion was M2 = 5.41%.

The visualization results of emotional resonance in short videos based on the micro cube model are shown in Fig 9.

### 4.3 Visualization results of scene transition of short videos based on microcube model

Based on the display data, the number of microcube models containing scene transitions and the number of video pumping seconds frames were recorded as shown in the Table 4.

**Table 3. The number of microcube models containing emotional elements and the video pumping second frames.**

|  | Popular city image short video | Non-popular city image short video |
|---|---|---|
| Number of microcube models containing emotional elements | 69 | 35 |
| Color matching of total model M1 | 47.92% | 28.46% |
| Color matching of total emotion M2 | 9.21% | 5.41% |
| Number of microcube models involving positive emotions | 47 | 32 |
| Color matching of total model M1 | 32.64% | 26.02% |
| Color matching of total emotion M2 | 61.84% | 91.43% |
| Number of microcubes models involving negative emotions | 29 | 5 |
| Color matching of total model M1 | 20.14% | 4.07% |
| Color matching of total emotion M2 | 38.16% | 13.51% |
| Number of microcubes models involving both positive and negative emotions | 7 | 2 |
| Color matching of total model M1 | 4.86% | 1.63 |
| Color matching of total emotion M2 | 9.21% | 5.41% |

The number of microcube models containing element q in popular city image videos is 91, accounting for 67.36%. among them, the number of microcube models with q = 2 is 14. the number of microcube models with q = 3 is 18. the number of microcube models with q = 4 is 26. the number of microcube models with q = 5 is 13. the number of microcube models with q = 6 is 8. the number of microcube models with q = 7 is The number of microcubes is 8. the number of microcubes with q = 8 is 3.

The number of microcube models containing element q in the non-popular city image videos is 26, accounting for 21.14%. among them, the number of microcube models with q = 2 is 3. the number of microcube models with q = 3 is 7. the number of microcube models with q = 4 is 6. the number of microcube models with q = 5 is 5. the number of microcube models with q = 6 is 1. the number of microcube models with q = 7 is 2. The number of models is 2.

The visualization results of scene transition of short videos based on microcube model are shown in Fig 10.

## 4.4 Visualization results of elemental amplification based on microcube model

Based on the display data, the number of microcube models containing the six types of imagery elements and the number of video pumping seconds frames were recorded as shown in the Table 5.

The number of microcubes containing imagery elements in popular city image videos is 141, the number of microcubes containing one type of imagery elements is 93, and the number of microcubes containing two or more types of imagery elements is 48. Among them, the number of microcubes involving technology elements is 3, the number of microcubes involving food elements is 33, the number of microcubes involving natural landscape elements is 21, the number of microcubes involving urban landscape elements is 74, the number of microcubes involving people elements is 66, and the number of microcubes involving animal elements is 4.

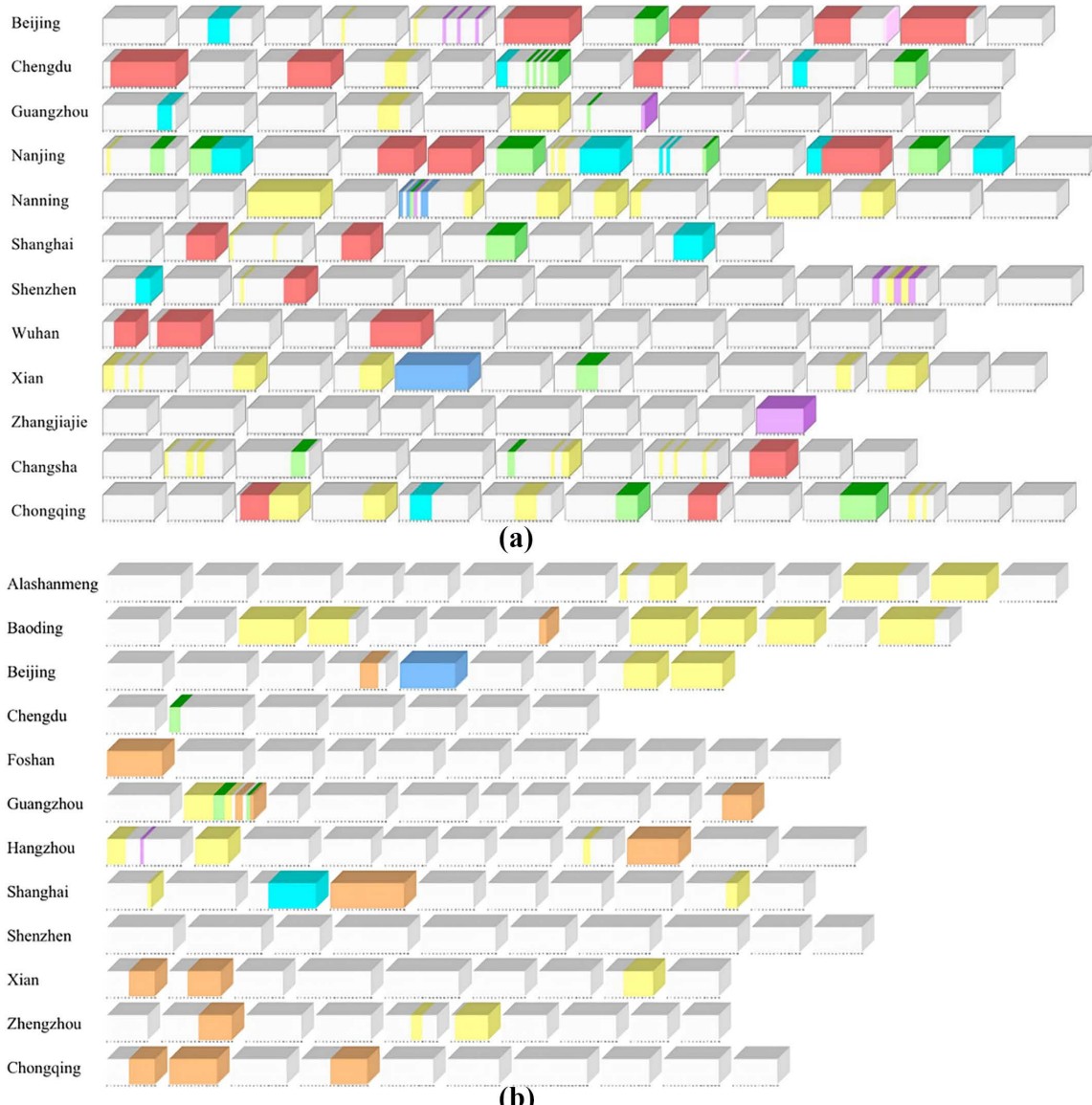

**Fig 9. Emotional resonance stacking diagram based on microcube model** (a) sample case of popular city image short video. (b) sample case of non-popular city image short video.

**Table 4. Number and percentage of microcube models containing scene transformation.**

|  | Popular city image short video | Non-popular city image short video |
|---|---|---|
| Number/proportion of microcube models containing element q | 91/67.36% | 26/21.14% |
| Number of microcube models with q = 2/percentage | 14/10.37% | 3/2.44% |
| Number of microcube models with q = 3/percentage | 18/13.33% | 7/5.69% |
| Number of microcube models with q = 4/percentage | 26/19.26% | 6/4.88% |
| Number of microcube models with q = 5/percentage | 13/9.63% | 5/4.07% |
| Number of microcube models with q = 6/percentage | 8/5.93% | 1/0.81% |
| Number of microcube models with q = 7/percentage | 8/5.93% | 2/1.63% |
| Number of microcube models with q = 8/percentage | 3/2.22% | 0/0% |

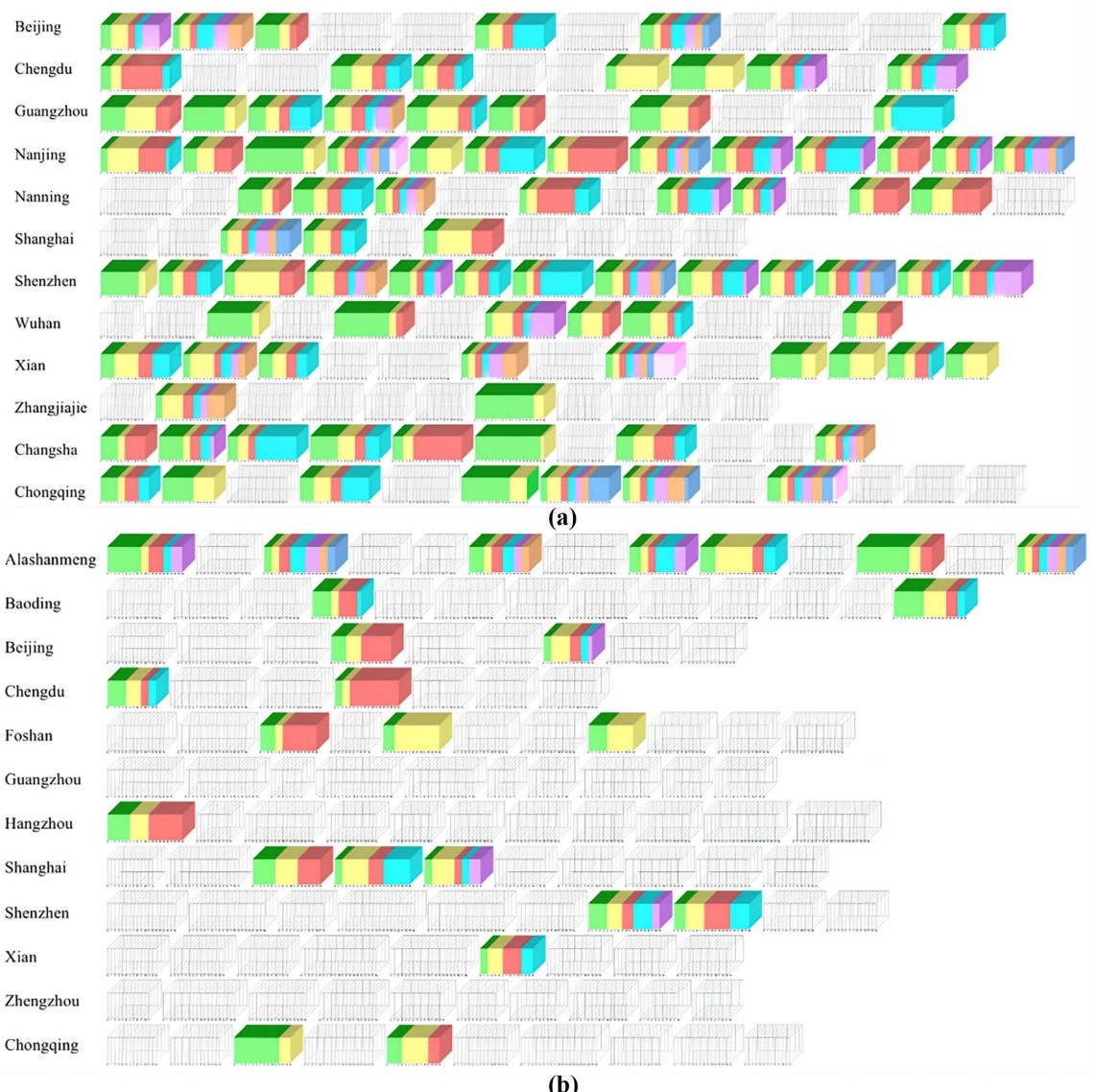

**Fig 10. Scene transformation stacking diagram based on microcube mode** (a) sample case of popular city image short video. (b) sample case of non-popular city image short video.

**Table 5. Number of microcube models containing intentional elements.**

| | Popular city image short video | Non-popular city image short video |
|---|---|---|
| Number of microcubes containing imagery elements | 141 | 116 |
| Number of microcubes with one type of imagery | 93 | 104 |
| Number of microcubes with two or more types of imagery | 48 | 12 |
| Number of microcubes with technological elements | 3 | 0 |
| Number of microcubes with food elements | 33 | 0 |
| Number of microcubes with natural landscape elements | 21 | 6 |
| Number of microcubes with urban landscape elements | 74 | 25 |
| Number of microcubes with human elements | 66 | 89 |
| Number of microcubes with animal elements | 4 | 0 |

The number of microcubes containing imagery elements in the non-popular city image videos is 116, the number of microcubes containing one type of imagery elements is 104, and the number of microcubes containing two or more types of imagery elements is 12. Among them, the number of microcubes for urban landscape elements is 25, the number of microcubes for people elements is 89, and the number of microcubes for natural landscape elements is 6.

The visualization results of elemental amplification based on microcube model are shown in Fig 11.

## 4.5  Visualization results of element interaction of short movies based on microcube model

Based on the display data of element interactions, the number of microcube models containing six types of imagery element interactions and the number of video pumping second frames were recorded as shown in the following Table 6.

The number of microcubes containing interactive elements in the popular city image videos is 17, and the number of video frames is 150. Among them, the number of microcubes with dynamic interactive elements equal to 2 is 10, and the number of video frames is 96. the number of microcubes with dynamic interactive elements greater than 2 is 7, and the number of video frames is 54.

The number of microcubes containing interactive elements in the non-popular city image videos is 9, and the number of video frames is 89. Among them, the number of microcubes with dynamic interactive elements of a certain type is equal to 2 is 8, and the number of video frames is 67. the number of microcubes with dynamic interactive elements of a certain type is greater than 2 is 2, and the number of video frames is 22.

The visualization results of element interaction of short movies based on microcube model are shown in Fig 12.

## 4.6  Visualization results of short video screen style based on the microcube model

Based on the display data, the number of microcube models containing 5 types of hues and the number of video pumping seconds frames were recorded as shown in the Table 7.

The number of microcubes with gray tones in the popular city image videos is 31, and the number of video frames is 75. the number of microcubes with dark tones is 43, and the number of video frames is 391. the number of microcubes with bright tones is 42, and the number of video frames is 369. the number of microcubes with medium tones is 74, and the number of video frames is 931. and the number of microcubes with light tones is 47, and the number of video frames is 429. The number of microcube models in lighter tones is 47 and the number of video frames is 429.

The non-popular city image videos contain 25 gray-toned microcubes and 26 video frames, 11 dark-toned microcubes and 128 video frames, 42 bright-toned microcubes and 556 video frames, 31 medium-toned microcubes and 452 video frames, and light-toned microcubes and video frames. The number of light-toned microcubes is 31 and the number of video frames is 429.

The visualization results of short video screen style based on the microcube model are shown in Fig 13.

## 6  Conclusion and discussion

This study proposes a microcube model to analyze the relationship between short video content characteristics and the effectiveness of city image promotion on the Douyin platform. Through a combination of quantitative analysis and visualization techniques, the research identifies key content factors, including unexpected events, emotional resonance, scene

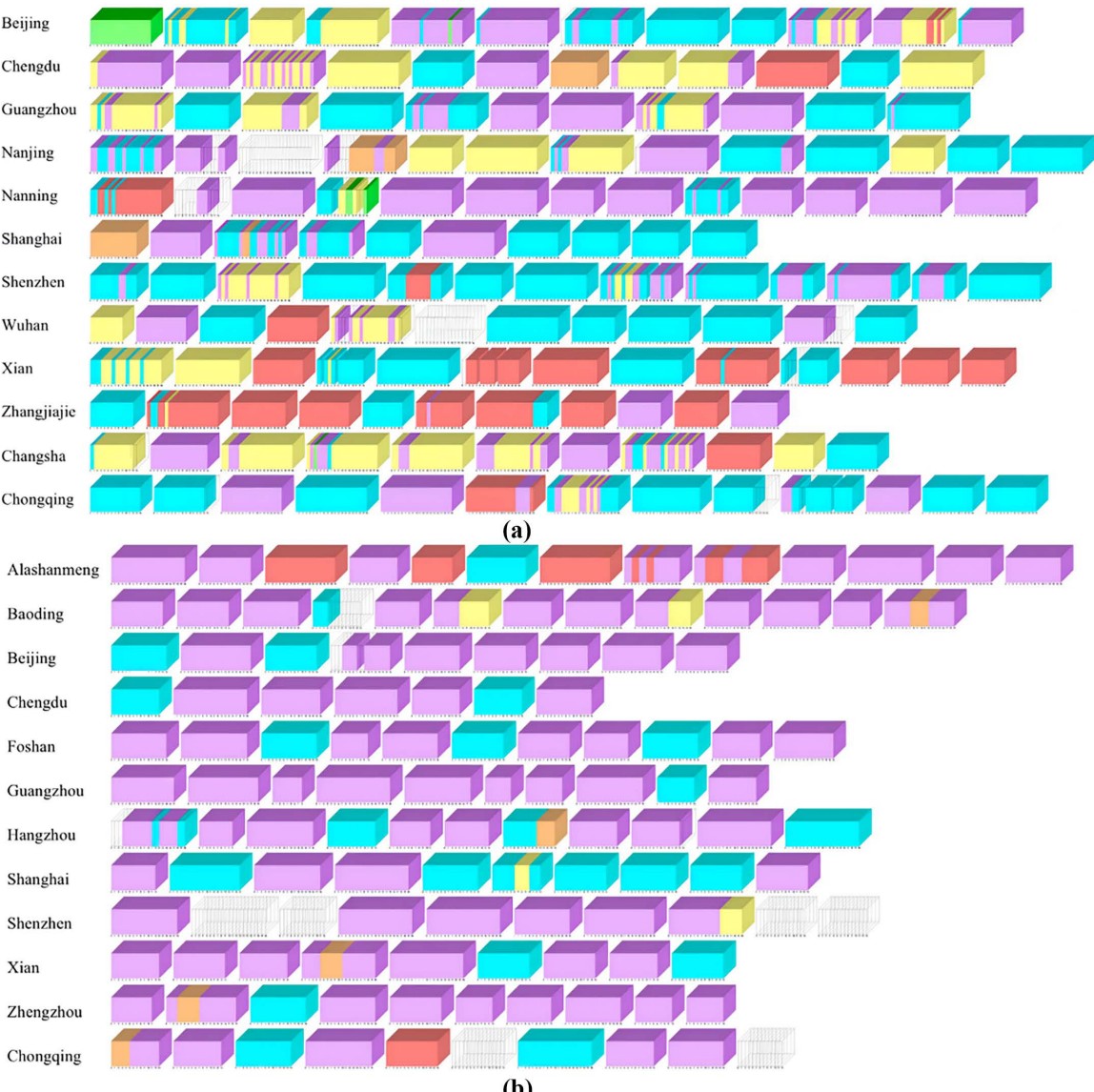

**Fig 11. Amplified stacking diagram of elements based on microcube model** (a) sample case of popular city image short video. (b) sample case of non-popular city image short video.

**Table 6. Number of microcube models containing element interactions and number of video pumping seconds frames.**

| | Popular city image short video | Non-popular city image short video |
|---|---|---|
| Number of microcube models with element interaction | 17 | 9 |
| Video draw seconds frames (sheets) | 150 | 89 |
| Number of microcube models with dynamic interaction of elements of a certain type of imagery = 2 | 10 | 8 |
| Video second draws (sheets) | 96 | 67 |
| Number of microcube models with dynamic interactive subjects > 2 for a certain type of imagery elements | 7 | 2 |
| Video second draws (sheets) | 54 | 22 |

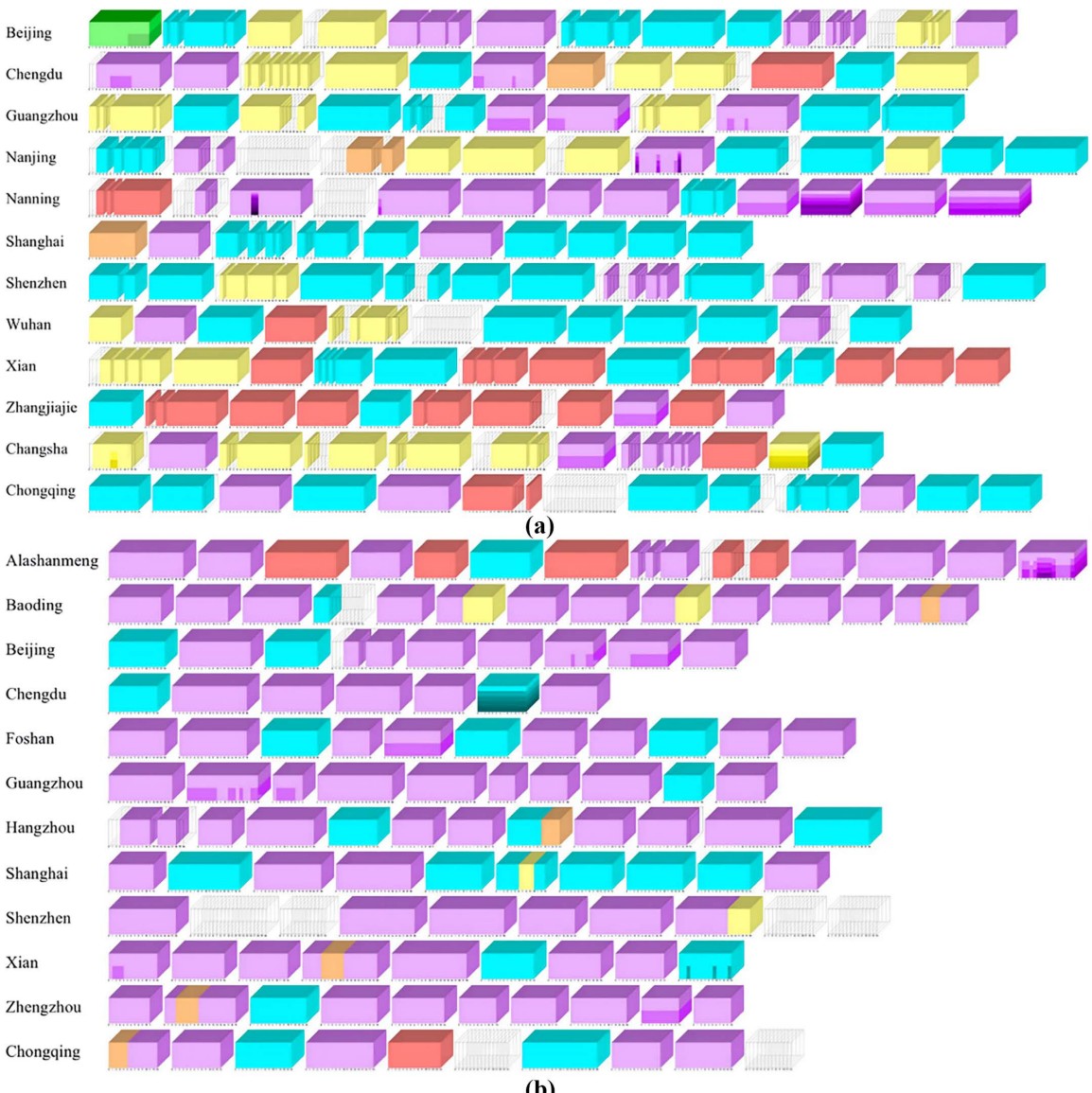

**Fig 12. Element interaction stacking diagram based on microcube model** (a) sample case of popular city image short video. (b) sample case of non-popular city image short video.

**Table 7. Number of microcube models containing unexpected events and number of video pumping seconds frames.**

|  | Popular city image short video | Non-popular city image short video |
|---|---|---|
| Number of microcube models with gray tones | 31 | 25 |
| Number of video frames in seconds (sheets) | 75 | 26 |
| Number of microcube models containing dark tones | 43 | 11 |
| Video frames (sheets) | 391 | 128 |
| Number of microcube models with bright tones | 42 | 42 |
| Video frames (pictures) | 369 | 556 |
| Number of microcube models with mid-tone colors | 74 | 31 |
| Video frames (pictures) | 931 | 452 |
| Number of microcubes with light tones | 47 | 31 |
| Video frames (sheets) | 429 | 428 |

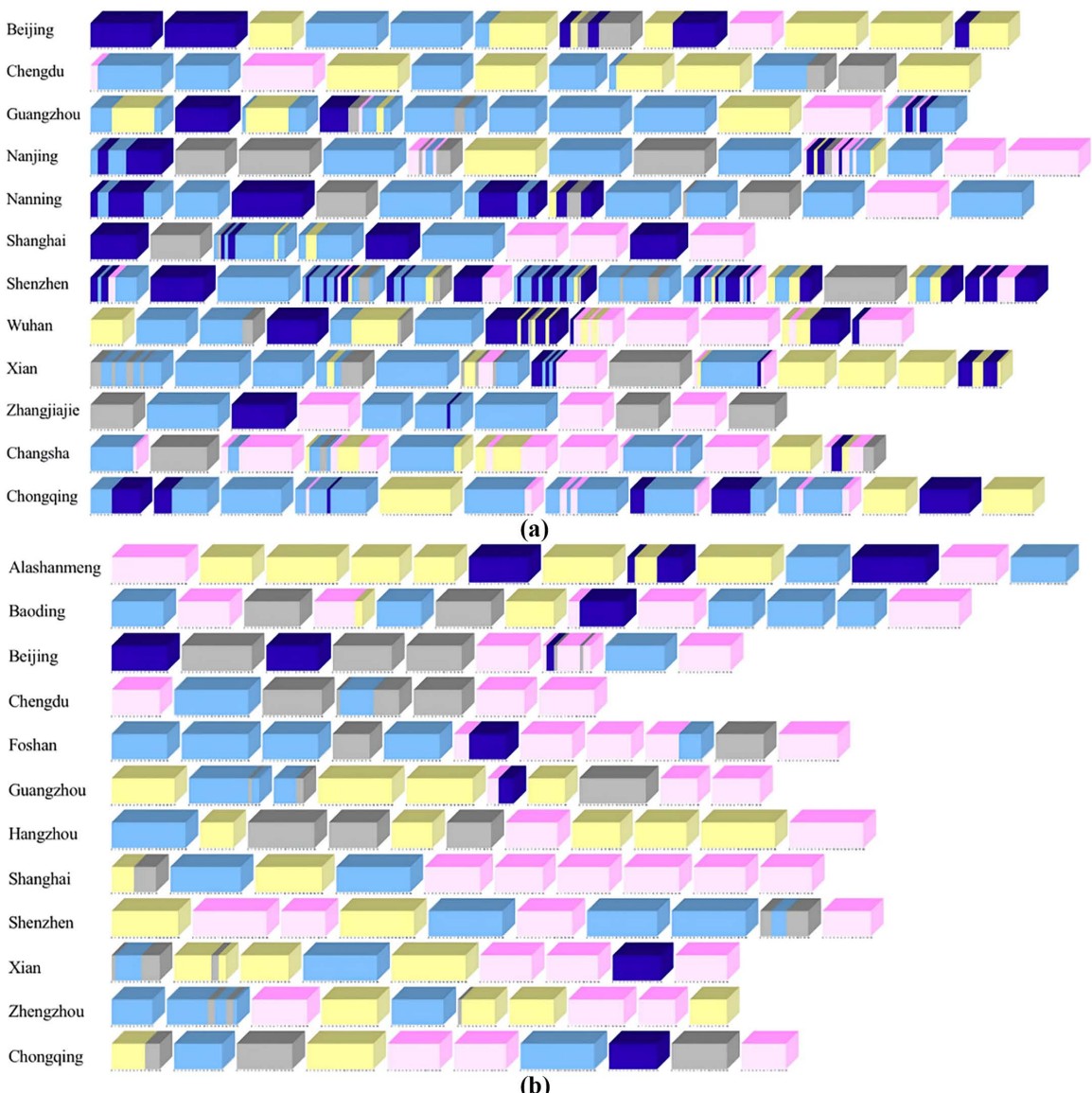

**Fig 13. Screen style stacking diagram based on microcube model** (a) sample case of popular city **image short video.** (b) sample case of non-popular city image short video.

transitions, element amplification, element interaction, and screen style, that significantly influence audience engagement and communication outcomes. The findings reveal that unexpected events capture attention effectively, while emotional resonance, particularly positive emotions, enhances the likelihood of sharing and deeper audience connection. Additionally, smooth scene transitions improve narrative coherence, and amplified visual elements combined with multi-element interactions create more engaging and memorable content. Furthermore, consistent and aesthetically pleasing screen styles reinforce emotional attachment and improve overall audience perception. These insights provide both theoretical and practical contributions, offering a structured framework for understanding short video dissemination and actionable strategies for optimizing content design in urban branding efforts.

However, this study is not without its limitations. First, the dataset was collected exclusively from the Douyin platform, limiting the generalizability of the findings to other platforms like Instagram or YouTube. Second, the research focuses on content analysis, lacking behavioral data such as click-through rates, viewing durations, and user interaction patterns. Third, the study's data collection period (July 2019 to December 2019) may not fully capture seasonal or temporal variations in content dissemination trends. Additionally, cultural and regional differences may influence how audiences respond to video content, limiting the findings' broader applicability.

Despite these limitations, this study introduces several innovative contributions. It develops a microcube model to provide a granular and dynamic method for analyzing short video data, establishes a multi-dimensional analytical framework to explore the relationship between content features and communication outcomes, and integrates quantitative and qualitative methods for a more comprehensive understanding of content dissemination dynamics. These contributions bridge the gap between theory and practice, offering city branding professionals and content creators valuable insights for designing impactful short video campaigns. Future research can address current limitations by expanding data sources across multiple platforms, incorporating audience behavior metrics, conducting cross-cultural comparisons, and exploring longitudinal trends to refine and validate the microcube model further. Through these efforts, cities can more effectively harness short video platforms to communicate their unique identities and engage global audiences.

## Author contributions

**Conceptualization:** Jing He.

**Formal analysis:** Zhuoluo Yang.

**Methodology:** Jiayi Zhu.

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
