## [Decision Letter · Decision Letter 0]

30 Oct 2024

PONE-D-24-25058A framework for visualizing and describing city image promotion short video data based on microcube modelPLOS ONE

Dear Dr. He,

Thank you for submitting your manuscript to PLOS ONE. After careful consideration, we feel that it has merit but does not fully meet PLOS ONE’s publication criteria as it currently stands. Therefore, we invite you to submit a revised version of the manuscript that addresses the points raised during the review process.

**ACADEMIC EDITOR: ** The reviewers' feedback has been received. Please make substantial revisions based on their suggestions.

We look forward to receiving your revised manuscript.

Kind regards,

Chao Gu

Academic Editor

PLOS ONE

3. In your Methods section, please include additional information about your dataset and ensure that you have included a statement specifying whether the collection and analysis method complied with the terms and conditions for the source of the data.

4. Thank you for stating the following financial disclosure: [This study was supported by Research Fund of Guangxi Key Lab of Multi-source Information Mining & Security (No.MIMS22-11), Key Laboratory of Spatial Data Mining & Information Sharing of Ministry of Education, Fuzhou University (No.2023LSDMIS02).]. Please state what role the funders took in the study. If the funders had no role, please state: "The funders had no role in study design, data collection and analysis, decision to publish, or preparation of the manuscript." If this statement is not correct you must amend it as needed. Please include this amended Role of Funder statement in your cover letter; we will change the online submission form on your behalf.

5. In the online submission form, you indicated that [Data is available upon reasonable request.]. All PLOS journals now require all data underlying the findings described in their manuscript to be freely available to other researchers, either 1. In a public repository, 2. Within the manuscript itself, or 3. Uploaded as supplementary information. This policy applies to all data except where public deposition would breach compliance with the protocol approved by your research ethics board. If your data cannot be made publicly available for ethical or legal reasons (e.g., public availability would compromise patient privacy), please explain your reasons on resubmission and your exemption request will be escalated for approval.

Reviewers' comments:

Reviewer's Responses to Questions

**Comments to the Author**

1. Is the manuscript technically sound, and do the data support the conclusions?

Reviewer #1: Partly

Reviewer #2: Yes

Reviewer #3: Partly

Reviewer #4: Partly

2. Has the statistical analysis been performed appropriately and rigorously? 

Reviewer #1: Yes

Reviewer #2: Yes

Reviewer #3: No

Reviewer #4: I Don't Know

3. Have the authors made all data underlying the findings in their manuscript fully available?

Reviewer #1: No

Reviewer #2: Yes

Reviewer #3: Yes

Reviewer #4: No

4. Is the manuscript presented in an intelligible fashion and written in standard English?

Reviewer #1: Yes

Reviewer #2: Yes

Reviewer #3: Yes

Reviewer #4: Yes

5. Review Comments to the Author

Reviewer #1: Authors attempt to work on a very interesting topic regarding showcasing cities for the purpose of popularity and prosperity however the real problem is not clear as to why this work is necessary.

It is recommended that authors should remove the sections indicated in abstract but only retain the logical structure. Also, the introduction section looks shallow and the background to this study requires more robust introduction section reflecting the real problem with global relevance for proper perspective.

It is important to reflect on how such framework could affect tourism either positively or negatively due to the fact that some cities are dream land to many globally and the moment they view such videos it could impact their decisions in both ways hence the framework must consider this effect.

The source of the TikTok video is missing even though authors stated that they were verified. Where were this data stored for the purpose of this research among other data integrity concerns.

While commending the analysis of cases and extensive review carried out by authors figures e.g. figure 7 is not clear enough to relate with as much as readership requires great improvement. It is recommended that authors should device an alternative means of representing this fact.

For clarity it is important for the authors to discuss results separately from conclusion as it looks mixed up and vague thereby making it difficult to see in clear terms what achievement is recorded by this work.

Grammatical flow should be checked by authors for consistency

Reviewer #2: A framework for visualizing and describing city image promotion short video data based on microcube model.

Reviewed by: Riyaz Abro.

Observations (If appropriate & Applicable).

• Abstract:

o Which software was used to analyse the data, include timeframe and time limit of the data that from what time and period the data was collected.

• Introduction:

o Already given variables in keywords are different than the defined ones

• Literature Review:

o Given variables must be defined (if possible).

• Precise Conclusion:

o Limitation must be defined, hypotheses must be generated separately, however, paper is very much innovative and be published with minor correction as suggested and if deemed appropriate.

Reviewer #3: 1. Explain the immersion methos or techniques based on micro cube model

2. Also explain how all the parameters are interrelated with each other

3.One absence of parameter will affect the performance of the short videos

Reviewer #4: This paper uses the Microcube model framework to calculate the unit video content and analyze the packaging time in each microcube, which provides a novel perspective for city image promotion and has certain research value, but there are still the following shortcomings, which I hope to improve:

1. Literature review part of the literature, mainly focused on China literature research, the lack of a broader field and country, the review of relevant research results. In view of the globalization background of short video and urban communication and promotion, the research results of international scholars in the fields of short video content and communication mode should be introduced appropriately. This will help to enhance the global perspective and academic depth of this study, and more comprehensively demonstrate the applicability and innovation of the study on a global scale.

2. Short videos are divided into six categories, but no specific classification basis is provided. The current classification may not fully cover the diversity of short videos, especially when the differences in content and communication targets increase. It is suggested to provide clearer classification standards and consider how to incorporate other types of video content into the classification system to enhance the scientificity and applicability of classification.

3. The correspondence between color and concept needs to be explained: the study used color to distinguish different elements when discussing short video content, but did not explain in detail how the color corresponds to specific elements.

4. There is a lack of transparency in the process of constructing the micro-cube model. The micro-cube model is one of the core methods in this study, but at present, the description of the model construction process is abstract, and the existing algorithm (data slicing? Or is it an optimization algorithm for slicing? Or OLAP and multidimensional data modeling? ) or self-construction algorithm, the paper does not explain in detail, and the listed algorithms lack detailed explanation. In order to improve the transparency and reproducibility of the model, I hope the author can explain it in detail and provide specific procedures, formulas and data support for the model construction. This will facilitate colleagues to better understand and verify the operability and effectiveness of the model.

5. Scientific methods in the process of short video classification: At present, short video classification methods lack systematic scientific basis and rely on subjective judgment, which may affect the accuracy of classification and the rigor of research results. It is suggested that a more systematic classification method should be introduced in the classification process.

6. PLOS authors have the option to publish the peer review history of their article (what does this mean? ). If published, this will include your full peer review and any attached files.

**Do you want your identity to be public for this peer review?** For information about this choice, including consent withdrawal, please see our Privacy Policy .

Reviewer #1: No

Reviewer #2: **Yes: ** Riyaz Abro

Reviewer #3: **Yes: ** Dr. Alankrita Aggarwal

Reviewer #4: No

---

## [Author Response · Author response to Decision Letter 0]

17 Nov 2024

Review Comments to the Author

Reviewer #1:

Authors attempt to work on a very interesting topic regarding showcasing cities for the purpose of popularity and prosperity however the real problem is not clear as to why this work is necessary.

Response: Thank the reviewers for their careful reading and valuable comments on this article. We recognize that there is room for improvement in clarifying the context and necessity of the study. In the process of revision, we re-sorted out the problem consciousness and goal of the research, further supplemented the specific description of the research background and the potential application value of this research in the analysis of short video content of urban image promotion.

It is recommended that authors should remove the sections indicated in abstract but only retain the logical structure. Also, the introduction section looks shallow and the background to this study requires more robust introduction section reflecting the real problem with global relevance for proper perspective.

Response: Thanks to the reviewers for their pertinent suggestions on this paper, especially for the guidance on simplifying the content of the abstract and perfecting the background of the introduction. We recognize that the current summary is somewhat redundant in the description of technical details, which may reduce the clarity of the core logical structure. According to the recommendations of the reviewers, we have deleted technical expressions in the revision and highlighted the logical thread in order to more directly demonstrate the research objectives and significance. In addition, as for the problems in the introduction, we are deeply aware that the original text lacks in the breadth and depth of the research background, especially in the context of globalization, the expression of relevance still needs to be strengthened. To this end, we supplement the status quo and challenges of global urban brand communication, and combine the communication characteristics of short video, a new medium, to clarify the practical significance and scope of application of this study. It is hoped that these modifications can better reflect the overall logic and research value of the paper.

It is important to reflect on how such framework could affect tourism either positively or negatively due to the fact that some cities are dream land to many globally and the moment they view such videos it could impact their decisions in both ways hence the framework must consider this effect.

Response: Thanks to the reviewers for their suggestions on this paper, especially the concern that the research framework may influence travel decisions. We recognize that in the context of globalization, the potential positive and negative effects of short video content on the tourism industry deserve further discussion. In the revision, we have added relevant background information in the introduction, clearly pointing out the two-way possibilities of short videos in attracting tourists and affecting the image of the city. At the same time, in the discussion part, the specific analysis of the influence of the framework on tourism decision-making is added, including the possibility of positive and negative effects and countermeasures.

The source of the TikTok video is missing even though authors stated that they were verified. Where were this data stored for the purpose of this research among other data integrity concerns.

Response: Thank the reviewers for their attention to the source and completeness of the data in this paper. These issues are crucial to the credibility of the research, and we have made the following improvements in the revision: In the part of methodology, we detailed the collection and verification process of TikTok video data, including screening criteria, verification mechanism and denoising methods, to ensure the authenticity and representativeness of the data, and added specific measures for data storage and integrity protection.

While commending the analysis of cases and extensive review carried out by authors figures e.g. figure 7 is not clear enough to relate with as much as readership requires great improvement. It is recommended that authors should device an alternative means of representing this fact.

Response: Figure 7 has been reuploaded and improved to clearly label the variable meanings of each column and row, while ensuring good readability in both digital and print media by increasing color contrast and labeling.

For clarity it is important for the authors to discuss results separately from conclusion as it looks mixed up and vague thereby making it difficult to see in clear terms what achievement is recorded by this work.

Response: Thanks to the reviewers for their meticulous feedback on the structure of this paper. We recognize that mixing "results" with "conclusions" can affect readers' understanding of research findings and summaries. To this end, during the revision process, we reorganized the paper to include a separate "results" section to elaborate on specific findings based on the data analysis; At the same time, the "conclusion" part is adjusted to summarize the significance, practical application, limitations and future direction of the research.

Grammatical flow should be checked by authors for consistency.

Response: Thanks to the reviewers for their valuable comments on the grammatical coherence of this paper. We are deeply aware of the importance of grammatical accuracy and fluency in the presentation of academic articles. Based on your suggestions, we have made checked by authors for consistency to the full text.

Reviewer #2:

A framework for visualizing and describing city image promotion short video data based on microcube model.

Reviewed by: Riyaz Abro.

Observations (If appropriate & Applicable).

• Abstract:

o Which software was used to analyse the data, include timeframe and time limit of the data that from what time and period the data was collected.

Response: Thanks to the reviewers for their valuable suggestions on the summary. We recognize that clarity about the software tools used for data analysis and the time frame of the data helps improve transparency and reproducibility. In response to your suggestions, we have supplemented the data sources and research methods in the Summary and Methods section. Regarding the software used for data analysis, we utilized Python with pandas for data manipulation, scikit-learn for statistical analysis, and matplotlib for visualization. Additionally, deep learning techniques based on PyTorch were employed. Detailed descriptions related to the data are provided in Chapter 4 and Table 1.

• Introduction:

o Already given variables in keywords are different than the defined ones

Response: "City image publicity" has been changed to "city image promotion", and "microscopic content analysis" have been modified to "Micro Perspective".

• Literature Review:

o Given variables must be defined (if possible).

Response: We have revised the manuscript to include precise definitions of the key variables, such as "unexpected events," "emotional resonance," "scene transition," "elemental amplification," "element interaction," and "screen style." These definitions have been integrated into the literature review section to enhance clarity and provide a better understanding of their role in our research framework.

• Precise Conclusion:

o Limitation must be defined, hypotheses must be generated separately, however, paper is very much innovative and be published with minor correction as suggested and if deemed appropriate.

Response: Thank you very much for your insightful comments regarding the need for a precise conclusion. Based on your suggestions, we have further refined the "Conclusion and Discussion" section to clearly define the limitations of our study and have also generated the hypotheses separately. By doing so, we aim to provide a more structured and clear presentation of our research contributions.

Reviewer #3:

1. Explain the immersion methos or techniques based on micro cube model

Response: The immersion method based on the microcube model aims to analyze and enhance the viewer's immersive experience by breaking down short video content into multiple small content cube units. This model refines the video content by segmenting, assigning, and visualizing key elements within the short video, thereby exploring the impact of these elements on audience emotions and engagement.

2. Also explain how all the parameters are interrelated with each other

Response: The parameters identified in this study—unexpected events, emotional resonance, scene transition, element amplification, element interaction, and screen style—are deeply interrelated, working together to enhance the effectiveness of short videos in promoting city images.

A video that starts with an unexpected event can use emotional resonance to keep viewers engaged, turning their initial curiosity into a deeper emotional connection. Effective use of scene transitions combined with amplified elements can enhance the narrative clarity and visual impact, making the content more memorable. A well-defined screen style enhances element interactions by creating a cohesive visual experience that resonates with viewers, making the city image more distinctive. Each variable influences the others. For example, an unexpected event can be more impactful if accompanied by an emotionally resonant scene, amplified elements, and smooth transitions. Together, these parameters create a holistic effect, making the short video more engaging, memorable, and effective in conveying a city’s image. The absence of any one variable can reduce the overall effectiveness, as each plays a crucial role in maintaining viewer interest and enhancing the video’s communicative power.

3.One absence of parameter will affect the performance of the short videos

Response: Yes, the absence of any parameter will impact the overall performance of the short video. Including unexpected events in short videos can spark curiosity and quickly capture the audience's interest. Emotional resonance is a key factor in maintaining audience engagement, as emotion-driven content is critical for successful video dissemination. Proper scene transitions help maintain the video's pacing and narrative coherence, allowing viewers to immerse themselves more easily. Element amplification aims to capture viewers' attention through visual impact. Element interaction enhances the video's fun and dynamic presentation, making it more engaging and livelier. A unique screen style helps a video stand out among numerous others, creating a distinct memory point.

Reviewer #4:

This paper uses the Microcube model framework to calculate the unit video content and analyze the packaging time in each microcube, which provides a novel perspective for city image promotion and has certain research value, but there are still the following shortcomings, which I hope to improve:

1. Literature review part of the literature, mainly focused on China literature research, the lack of a broader field and country, the review of relevant research results. In view of the globalization background of short video and urban communication and promotion, the research results of international scholars in the fields of short video content and communication mode should be introduced appropriately. This will help to enhance the global perspective and academic depth of this study, and more comprehensively demonstrate the applicability and innovation of the study on a global scale.

Response: Thanks to the reviewers for their suggestions on the improvement of relevant work parts of this paper. We recognize that the addition of new academic research to each research topic not only improves the depth of the literature review, but also enhances the academic and theoretical support of the paper. According to your suggestions, we have added references from relevant global scholars' research results in six dimensions, namely, accident, emotional resonance, scene switching, element amplification, element interaction, and picture style.

2. Short videos are divided into six categories, but no specific classification basis is provided. The current classification may not fully cover the diversity of short videos, especially when the differences in content and communication targets increase. It is suggested to provide clearer classification standards and consider how to incorporate other types of video content into the classification system to enhance the scientificity and applicability of classification.

Response: We have revised the manuscript and provided detailed definitions of the six imagery elements. These elements are not intended for classifying short videos but were selected due to their significant impact on short video content. Therefore, they serve as the core research dimensions in the analysis framework for city image promotion short videos. These elements were derived from a comprehensive analysis of existing literature and empirical data, allowing us to better reveal the influence of short video content characteristics on city image promotion.

3. The correspondence between color and concept needs to be explained: the study used color to distinguish different elements when discussing short video content, but did not explain in detail how the color corresponds to specific elements.

Response: In the manuscript, we described how specific elements are represented using colors. For example, experimentally, we designed an emotion grid chart with nine emotions: moved, praised, funny, happy, neutral, angry, sad, shocked, and helpless. We used #EE6363, #EEDFCC, #FFA54F, #FFEC8B, #FFFFF0, #00F5FF, #63B8FF, #00FF7F, and #CD69C9 as RGB color controls, respectively, as shown in Figure 2(a). We designed an imagery elements grid with six elements: technology elements, food elements, natural landscape elements, urban landscape elements, people elements, animal elements. We used #00FF7F, #FFEC8B, #EE6363, #00F5FF, #EE82EE, #FFA54F as RGB color controls, as shown in Figure 4(a). And a grid diagram of the main colors of the screen style was designed, and #0000FF, #BEBEBE, #FFFACD, #6495ED, and #FFC0CB were used as RGB color controls for dark, gray, light, medium, and bright tones, respectively, as shown in Figure 6(a).

4. There is a lack of transparency in the process of constructing the micro-cube model. The micro-cube model is one of the core methods in this study, but at present, the description of the model construction process is abstract, and the existing algorithm (data slicing? Or is it an optimization algorithm for slicing? Or OLAP and multidimensional data modeling? ) or self-construction algorithm, the paper does not explain in detail, and the listed algorithms lack detailed explanation. In order to improve the transparency and reproducibility of the model, I hope the author can explain it in detail and provide specific procedures, formulas and data support for the model construction. This will facilitate colleagues to better understand and verify the operability and effectiveness of the model.

Response: Thank you for the your suggestion. The model construction process has been addressed in the manuscript. Specifically, we adopt a micro perspective to refine the video content by establishing a content microcube model that utilizes data slicing techniques to segment the complete short video into multiple fragmented units based on time intervals. Each microcube encapsulates the content per unit of time, as illustrated in Figure 1(a).

5. Scientific methods in the process of short video classification: At present, short video classification methods lack systematic scientific basis and rely on subjective judgment, which may affect the accuracy of classification and the rigor of research results. It is suggested that a more systematic classification method should be introduced in the classific

---

## [Decision Letter · Decision Letter 1]

22 Dec 2024

PONE-D-24-25058R1A framework for visualizing and describing city image promotion short video data based on microcube modelPLOS ONE

Dear Dr. He,

Thank you for submitting your manuscript to PLOS ONE. After careful consideration, we feel that it has merit but does not fully meet PLOS ONE’s publication criteria as it currently stands. Therefore, we invite you to submit a revised version of the manuscript that addresses the points raised during the review process. Please submit your revised manuscript by Feb 05 2025 11:59PM. If you will need more time than this to complete your revisions, please reply to this message or contact the journal office at plosone@plos.org . Please include the following items when submitting your revised manuscript:

We look forward to receiving your revised manuscript.

Kind regards,

Chao Gu

Academic Editor

PLOS ONE

Reviewers' comments:

Reviewer's Responses to Questions

**Comments to the Author**

1. If the authors have adequately addressed your comments raised in a previous round of review and you feel that this manuscript is now acceptable for publication, you may indicate that here to bypass the “Comments to the Author” section, enter your conflict of interest statement in the “Confidential to Editor” section, and submit your "Accept" recommendation.

Reviewer #1: All comments have been addressed

Reviewer #2: All comments have been addressed

Reviewer #3: All comments have been addressed

Reviewer #4: (No Response)

2. Is the manuscript technically sound, and do the data support the conclusions?

Reviewer #1: Yes

Reviewer #2: Yes

Reviewer #3: Yes

Reviewer #4: Partly

3. Has the statistical analysis been performed appropriately and rigorously? 

Reviewer #1: N/A

Reviewer #2: Yes

Reviewer #3: Yes

Reviewer #4: I Don't Know

4. Have the authors made all data underlying the findings in their manuscript fully available?

Reviewer #1: Yes

Reviewer #2: Yes

Reviewer #3: Yes

Reviewer #4: No

5. Is the manuscript presented in an intelligible fashion and written in standard English?

Reviewer #1: Yes

Reviewer #2: Yes

Reviewer #3: Yes

Reviewer #4: Yes

6. Review Comments to the Author

Reviewer #1: The feedback is noted and full data could be made available to the editor. Authors have done well in addressing most of the concerns raised.

Reviewer #2: A framework for visualizing and describing city image promotion short video data based on microcube model.

Reviewed by: Riyaz Abro.

Observations (If appropriate & Applicable).

• Abstract:

o Which software was used to analyse the data, include timeframe and time limit of the data that from what time and period the data was collected.

• Introduction:

o Already given variables in keywords are different than the defined ones

• Literature Review:

o Given variables must be defined (if possible).

• Precise Conclusion:

o Limitation must be defined, hypotheses must be generated separately, however, paper is very much innovative and be published with minor correction as suggested and if deemed appropriate.

Reviewer #3: (No Response)

Reviewer #4: Partial Literature Review: The literature review section is somewhat superficial and does not sufficiently cover the current state of research on urban image promotion through short videos and the micro-cube model, both domestically and internationally. The range of selected references is narrow, failing to fully reflect the major research achievements and theoretical developments in this field. I recommend that the authors expand the literature review to include more comprehensive studies on urban image promotion through short videos, particularly focusing on theoretical and methodological discussions in this area. Furthermore, the literature on the application of the micro-cube model in other fields should also be reviewed to provide a more complete academic context for the study.

Insufficient Explanation of the Key Elements: The six key elements proposed as the core components of urban image promotion through short videos are not sufficiently justified in the manuscript. While the selection of these elements seems logical, there is a lack of adequate theoretical rationale and empirical support to demonstrate why these six elements are the most critical. I recommend that the authors further explore and reference relevant literature or empirical studies to support the importance of these elements in short video promotion. Additionally, the authors should clarify how these elements specifically influence the effectiveness of urban image communication.

Insufficient Details in the Micro-Cube Model Construction: The paper proposes a visual and quantitative research method, but the construction of the micro-cube model lacks clear and detailed presentation of the experimental data and model-building process. This makes the model construction process opaque and prevents other researchers from replicating or adopting this method in future studies. To improve the reproducibility of the research, I suggest that the authors provide a more detailed explanation of each step in the micro-cube model construction, including the sources of experimental data, data processing methods, specific algorithms used, and the implementation process. This would not only enhance the credibility of the research but also offer clearer methodological guidance for future studies.

7. PLOS authors have the option to publish the peer review history of their article (what does this mean? ). If published, this will include your full peer review and any attached files.

**Do you want your identity to be public for this peer review?** For information about this choice, including consent withdrawal, please see our Privacy Policy .

Reviewer #1: No

Reviewer #2: **Yes: ** Riyaz Abro

Reviewer #3: **Yes: ** Dr.Alankrita Aggarwal

Reviewer #4: No

---

## [Author Response · Author response to Decision Letter 1]

30 Dec 2024

Review Comments to the Author

Reviewer #1: The feedback is noted and full data could be made available to the editor. Authors have done well in addressing most of the concerns raised.

Response：Thank you very much for your kind and encouraging feedback. We are pleased to hear that our revisions have addressed most of your concerns. Your constructive comments have significantly improved the clarity, rigor, and depth of our work. We deeply appreciate your time, effort, and valuable insights throughout the review process.

Reviewer #2: A framework for visualizing and describing city image promotion short video data based on microcube model.

Reviewed by: Riyaz Abro.

Observations (If appropriate & Applicable).

• Abstract:

o Which software was used to analyse the data, include timeframe and time limit of the data that from what time and period the data was collected.

• Introduction:

o Already given variables in keywords are different than the defined ones

• Literature Review:

o Given variables must be defined (if possible).

• Precise Conclusion:

o Limitation must be defined, hypotheses must be generated separately, however, paper is very much innovative and be published with minor correction as suggested and if deemed appropriate.

Response：Thank you very much for your meticulous review and valuable suggestions on our manuscript. Your insightful comments regarding the inclusion of data analysis software, data timeframe, and collection period in the abstract, alignment of keywords with defined variables in the introduction, clear definitions of variables in the literature review, and the separation of hypotheses and limitations in the conclusion have been extremely helpful in improving the quality of our work. We have carefully addressed your feedback by specifying the software used (Python, Pandas, Matplotlib) and the data collection timeframe (July 2019 to December 2019) in the abstract. In the introduction, we have ensured consistency between the keywords and the defined variables. The literature review now includes clear definitions of the core variables, each supported by relevant references. In the conclusion, we have separately outlined the hypotheses, clarified the study's limitations, and highlighted its innovative contributions. Once again, thank you for your recognition of our research and your constructive guidance. We hope the revised manuscript meets your expectations and look forward to your further review and feedback.

Reviewer #3: (No Response)

Reviewer #4: Partial Literature Review: The literature review section is somewhat superficial and does not sufficiently cover the current state of research on urban image promotion through short videos and the micro-cube model, both domestically and internationally. The range of selected references is narrow, failing to fully reflect the major research achievements and theoretical developments in this field. I recommend that the authors expand the literature review to include more comprehensive studies on urban image promotion through short videos, particularly focusing on theoretical and methodological discussions in this area. Furthermore, the literature on the application of the micro-cube model in other fields should also be reviewed to provide a more complete academic context for the study.

Response：Thank you for your thoughtful and constructive feedback on our manuscript. We sincerely appreciate your detailed suggestions regarding the Literature Review section. Your comments have highlighted critical areas where our work can be improved, and we have carefully addressed them in our revisions.

Insufficient Explanation of the Key Elements: The six key elements proposed as the core components of urban image promotion through short videos are not sufficiently justified in the manuscript. While the selection of these elements seems logical, there is a lack of adequate theoretical rationale and empirical support to demonstrate why these six elements are the most critical. I recommend that the authors further explore and reference relevant literature or empirical studies to support the importance of these elements in short video promotion. Additionally, the authors should clarify how these elements specifically influence the effectiveness of urban image communication.

Response：Thank you for your constructive feedback and for highlighting the need for a more comprehensive theoretical and empirical rationale for the six key elements—unexpected events, emotional resonance, scene transition, elemental amplification, element interaction, and screen style—proposed in our study on urban image promotion through short videos. We have carefully addressed your comments and made substantial revisions to clarify and justify the selection and importance of these elements.

Insufficient Details in the Micro-Cube Model Construction: The paper proposes a visual and quantitative research method, but the construction of the micro-cube model lacks clear and detailed presentation of the experimental data and model-building process. This makes the model construction process opaque and prevents other researchers from replicating or adopting this method in future studies. To improve the reproducibility of the research, I suggest that the authors provide a more detailed explanation of each step in the micro-cube model construction, including the sources of experimental data, data processing methods, specific algorithms used, and the implementation process. This would not only enhance the credibility of the research but also offer clearer methodological guidance for future studies.

Response：Thank you for your thoughtful and constructive feedback regarding the lack of clarity and transparency in the construction of the Micro-Cube Model. We fully acknowledge the importance of presenting a clear, replicable, and well-documented methodology to ensure the robustness and reproducibility of our research. In response to your suggestion, we have significantly revised the relevant section to provide a comprehensive overview of the model-building process, covering data sources, processing methods, algorithmic logic, and implementation steps in a streamlined and coherent narrative.

The Micro-Cube Model was developed to provide a quantitative and visual analytical framework for understanding the relationship between short video content characteristics and their effectiveness in urban image promotion. The dataset, consisting of 20,668 video screenshots, was collected from the Douyin platform over a six-month period (July 2019 to December 2019). Video samples were carefully screened to ensure they included clear geographic tags and an explicit urban branding theme. Irrelevant, duplicate, or noise-laden samples were excluded to maintain data integrity. Data collection adhered to platform usage policies, and metadata such as upload date, hashtags, and account identity were cross-verified for accuracy.

The dataset was processed using Python, with Pandas and Matplotlib libraries applied for data cleaning, organization, and visualization. The video timeline was segmented into micro time units, each represented by a Micro-Cube, functioning as the fundamental analytical unit. These cubes encapsulate video content at a granular level, allowing us to isolate and measure six core visual elements: unexpected events, emotional resonance, scene transition, elemental amplification, element interaction, and screen style. Each element was assigned binary values (1 for presence, 0 for absence), and color-coding techniques were applied for visualization. For example, emotional resonance was represented using a nine-color emotional grid, while scene transitions were mapped using temporal and spatial segmentation rules. The degree of element interaction was calculated by identifying overlapping or dynamically connected visual elements within each Micro-Cube.

The algorithmic logic of the Micro-Cube Model includes systematic rules for identifying and quantifying these visual elements. For instance, unexpected events were detected by analyzing content anomalies across consecutive frames, while emotional resonance was assessed using color-matching ratios within emotional grids. Scene transitions were quantified based on abrupt shifts in temporal or spatial dimensions, and elemental amplification was measured by the percentage of screen area occupied by prominent visual details. The dynamic interactions between visual elements were decomposed into sub-categories using color-based sub-coding techniques, allowing us to visualize how elements interact over time. Additionally, screen style was analyzed through a classification of primary color tones (e.g., bright, dark, neutral), mapped across time intervals to understand their distribution and impact on audience perception.

The implementation process followed a sequential workflow: (1) Data segmentation into time-based units, (2) Assignment of binary codes to visual elements, (3) Visualization of Micro-Cube distributions using heatmaps and time-series charts, and (4) Quantitative analysis of relationships between these visual elements and audience engagement metrics, such as likes, comments, and shares. By overlaying these data layers within the Micro-Cube visualization framework, we were able to reveal correlations and patterns that drive the effectiveness of short video content in promoting urban imagery.

Finally, we ensured that the Micro-Cube Model remains replicable and adaptable for other researchers. The detailed methodological descriptions, coupled with transparent documentation of data sources, processing workflows, and algorithmic principles, enable other scholars to replicate, validate, and extend our approach across different datasets and research contexts. Furthermore, while our study focuses on urban image promotion, the Micro-Cube Model is inherently flexible and can be applied to areas such as brand communication analysis, cultural content dissemination, and digital media optimization.

We believe these revisions address your concerns by providing a coherent, transparent, and reproducible overview of the Micro-Cube Model. The enhanced clarity in the data sources, processing techniques, algorithmic implementation, and visualization workflows ensures a stronger methodological foundation for our research while offering clear guidance for future studies.

Thank you again for your valuable suggestions, which have greatly improved the methodological transparency of our paper. We look forward to your further feedback.

---

## [Decision Letter · Decision Letter 2]

8 Jan 2025

A framework for visualizing and describing city image promotion short video data based on microcube model

PONE-D-24-25058R2

Dear Dr. He,

We’re pleased to inform you that your manuscript has been judged scientifically suitable for publication and will be formally accepted for publication once it meets all outstanding technical requirements.

Kind regards,

Chao Gu

Academic Editor

PLOS ONE

Additional Editor Comments (optional):

Congratulations! I agree with the opinions of the four reviewers, and this paper is suitable for recommendation for publication.

Reviewers' comments:

Reviewer's Responses to Questions

**Comments to the Author**

1. If the authors have adequately addressed your comments raised in a previous round of review and you feel that this manuscript is now acceptable for publication, you may indicate that here to bypass the “Comments to the Author” section, enter your conflict of interest statement in the “Confidential to Editor” section, and submit your "Accept" recommendation.

Reviewer #4: (No Response)

2. Is the manuscript technically sound, and do the data support the conclusions?

Reviewer #4: (No Response)

3. Has the statistical analysis been performed appropriately and rigorously? 

Reviewer #4: (No Response)

4. Have the authors made all data underlying the findings in their manuscript fully available?

Reviewer #4: (No Response)

5. Is the manuscript presented in an intelligible fashion and written in standard English?

Reviewer #4: (No Response)

6. Review Comments to the Author

Reviewer #4: (No Response)

7. PLOS authors have the option to publish the peer review history of their article (what does this mean? ). If published, this will include your full peer review and any attached files.

**Do you want your identity to be public for this peer review?** For information about this choice, including consent withdrawal, please see our Privacy Policy .

Reviewer #4: No

---

## [Editor Report · Acceptance letter]

PONE-D-24-25058R2

PLOS ONE

Dear Dr. He,

I'm pleased to inform you that your manuscript has been deemed suitable for publication in PLOS ONE. Congratulations! Your manuscript is now being handed over to our production team.

Kind regards,

on behalf of

Dr. Chao Gu

Academic Editor

PLOS ONE